elefesciences.org

# A molecular tweezer antagonizes seminal amyloids and HIV infection

Edina Lump[1†], Laura M Castellano[2,3†], Christoph Meier[4], Janine Seeliger[5], Nelli Erwin[5], Benjamin Sperlich[5], Christina M Stürzel[1], Shariq Usmani[1], Rebecca M Hammond[2,6], Jens von Einem[7], Gisa Gerold[8], Florian Kreppel[9], Kenny Bravo-Rodriguez[10], Thomas Pietschmann[8], Veronica M Holmes[11], David Palesch[1], Onofrio Zirafi[1], Drew Weissman[11], Andrea Sowislok[12], Burkhard Wettig[12], Christian Heid[12], Frank Kirchhoff[1,13], Tanja Weil[4,13], Frank-Gerrit Klärner[14], Thomas Schrader[12], Gal Bitan[15,16,17], Elsa Sanchez-Garcia[10], Roland Winter[5], James Shorter[2,3*], Jan Münch[1,13*]

[1]Institute of Molecular Virology, Ulm University Medical Center, Ulm, Germany; [2]Department of Biochemistry and Biophysics, Perelman School of Medicine at the University of Pennsylvania, Philadelphia, United States; [3]Pharmacology Graduate Group, Perelman School of Medicine at the University of Pennsylvania, Philadelphia, United States; [4]Institute of Organic Chemistry III/Macromolecular Chemistry, Ulm University, Ulm, Germany; [5]Physical Chemistry I—Biophysical Chemistry, Department of Chemistry and Chemical Biology, Technical University of Dortmund, Dortmund, Germany; [6]Biology Department, Swarthmore College, Swarthmore, United States; [7]Institute of Virology, Ulm University Medical Center, Ulm, Germany; [8]Institute of Experimental Virology, Twincore, Centre for Experimental and Clinical Infection Research, Hannover, Germany; [9]Institute of Gene Therapy, Ulm University Medical Center, Ulm, Germany; [10]Max-Planck-Institut für Kohlenforschung, Mülheim an der Ruhr, Germany; [11]Division of Infectious Diseases, Department of Medicine, Perelman School of Medicine at the University of Pennsylvania, Philadelphia, United States; [12]Department of Chemistry, University of Duisburg-Essen, Essen, Germany; [13]Ulm-Peptide Pharmaceuticals, Ulm University, Ulm, Germany; [14]Institute of Organic Chemistry, University of Duisburg-Essen, Essen, Germany; [15]Department of Neurology, David Geffen School of Medicine, University of California, Los Angeles, Los Angeles, United States; [16]Brain Research Institute, University of California at Los Angeles, Los Angeles, Los Angeles, United States; [17]Molecular Biology Institute, University of California, Los Angeles, United States

*For correspondence: jshorter@mail.med.upenn.edu (JS); Jan.Muench@uni-ulm.de (JM)

†These authors contributed equally to this work

**Abstract** Semen is the main vector for HIV transmission and contains amyloid fibrils that enhance viral infection. Available microbicides that target viral components have proven largely ineffective in preventing sexual virus transmission. In this study, we establish that CLR01, a 'molecular tweezer' specific for lysine and arginine residues, inhibits the formation of infectivity-enhancing seminal amyloids and remodels preformed fibrils. Moreover, CLR01 abrogates semen-mediated enhancement of viral infection by preventing the formation of virion–amyloid complexes and by directly disrupting the membrane integrity of HIV and other enveloped viruses. We establish that CLR01 acts by binding to the target lysine and arginine residues rather than by a non-specific, colloidal mechanism. CLR01 counteracts both host factors that may be important for HIV transmission and the pathogen itself. These combined anti-amyloid and antiviral activities make CLR01 a promising topical microbicide for blocking infection by HIV and other sexually transmitted viruses.

**eLife digest** Human Immunodeficiency Virus (HIV) is a sexually transmitted virus that can cause a serious disease that weakens the immune system. The virus is most commonly transmitted between individuals in semen, the male reproductive fluid. Semen contains deposits of protein fragments called amyloid fibrils, which can increase the transmission of HIV by trapping viral particles. This helps the virus to attach to the membranes surrounding human cells, which increases the risk of infection. Therefore, therapies that reduce the levels of amyloid fibrils in semen might be able to reduce the transmission of HIV.

Drugs that prevent amyloid formation are already being developed because structurally similar fibrils can also form in the brains of individuals with neurodegenerative diseases. One such molecule—called CLR01—works by binding to particular sites on the proteins that form fibrils in the brain. This inhibits fibril formation and slowly disassembles the fibrils that have already formed. CLR01 physically interacts with these residues in a way that resembles a tweezer.

The peptides in the amyloid fibrils in semen also have these sites, which suggests that CLR01 might also disrupt amyloid fibrils from forming in semen. Here Lump and Castellano et al. show that CLR01 can both disrupt fibril formation and remodel fibrils that have already formed. In addition, CLR01 prevents HIV particles from interacting with these fibrils and can displace the virus particles that have already bound to the fibrils. In the presence of CLR01, human cells exposed to semen that contained HIV were less likely to become infected with the virus.

Unexpectedly, CLR01 also directly destroys HIV and other enveloped viruses such as HCV or HSV particles by disrupting the membranes that surround the virus. Therefore, Lump and Castellano et al.'s findings reveal that CLR01 has considerable potential to be used as an agent for reducing the transmission of HIV and other sexually transmitted viral diseases.

## Introduction

The majority of new HIV-1 infections are transmitted via sexual intercourse, and semen is the main vector for viral spread. Far from being a passive vehicle, semen potently enhances HIV infectivity (*Münch et al., 2007*; *Kim et al., 2010*). This HIV-enhancing activity is attributed to seminal amyloid fibrils (*Münch et al., 2007*; *Kim et al., 2010*; *Roan et al., 2011*; *Usmani et al., 2014*) that form by self-assembly of proteolytic fragments of prostatic acid phosphatase (PAP248-286 and PAP85-120) and the homologous proteins semenogelin 1 (SEM1) and semenogelin 2 (SEM2) (*Münch et al., 2007*; *Roan et al., 2011*; *Arnold et al., 2012*; *Castellano and Shorter, 2012*; *Münch et al., 2014*). Seminal amyloid fibrils are highly cationic, and the positively charged fibrils capture HIV virions, increase viral attachment rates to target cells, and augment fusion (*Roan et al., 2009*; *Arnold et al., 2012*; *Usmani et al., 2014*). By doing so, fibrils promote HIV infection in vitro by several orders of magnitude, whereas the corresponding monomeric peptides have no effect (*Münch et al., 2007*; *Roan et al., 2011*; *Arnold et al., 2012*). Importantly, the stimulatory effect of seminal amyloid is the greatest at low virus concentrations (*Münch et al., 2007*), and semen and PAP248-286 fibrils (termed SEVI for Semen-derived Enhancer of Virus Infection) may facilitate vaginal virus transmission after exposure to low viral doses (*Miller et al., 1994*; *Neildez et al., 1998*; *Münch et al., 2013*). HIV transmission rates are relatively low, occurring as infrequently as 1 in 200 to as low as 1 in 10,000 coital acts (*Gray et al., 2001*). Thus, counteraction of infectivity promoting amyloids in semen should reduce or even prevent HIV transmission via the sexual route.

The lysine- and arginine-specific molecular tweezer, CLR01 (*Figure 1A,B*) (*Fokkens et al., 2005*; *Klärner et al., 2006*, *2010*), inhibits amyloid fibrillization by engaging specific lysine, arginine, or both residues within a variety of disease-associated amyloidogenic proteins including amyloid-β protein (Aβ), tau, islet amyloid polypeptide, transthyretin, and α-synuclein (*Sinha et al., 2011*; *Attar et al., 2012*; *Prabhudesai et al., 2012*; *Sinha et al., 2012*; *Acharya et al., 2014*; *Ferreira et al., 2014*; *Lopes et al., 2015*; *Zheng et al., 2015*). Furthermore, CLR01 has even been found to slowly remodel preformed Aβ and α-synuclein fibrils over the course of several weeks (*Sinha et al., 2011*; *Prabhudesai et al., 2012*). CLR01 binds lysine residues with a $K_d$ of ~10 μM and also binds arginine residues, albeit with ~10-fold lower affinity (*Fokkens et al., 2005*; *Dutt et al., 2013*). The unprecedented high specificity of CLR01 for basic amino acids relies on a unique binding mode in which the tweezer draws

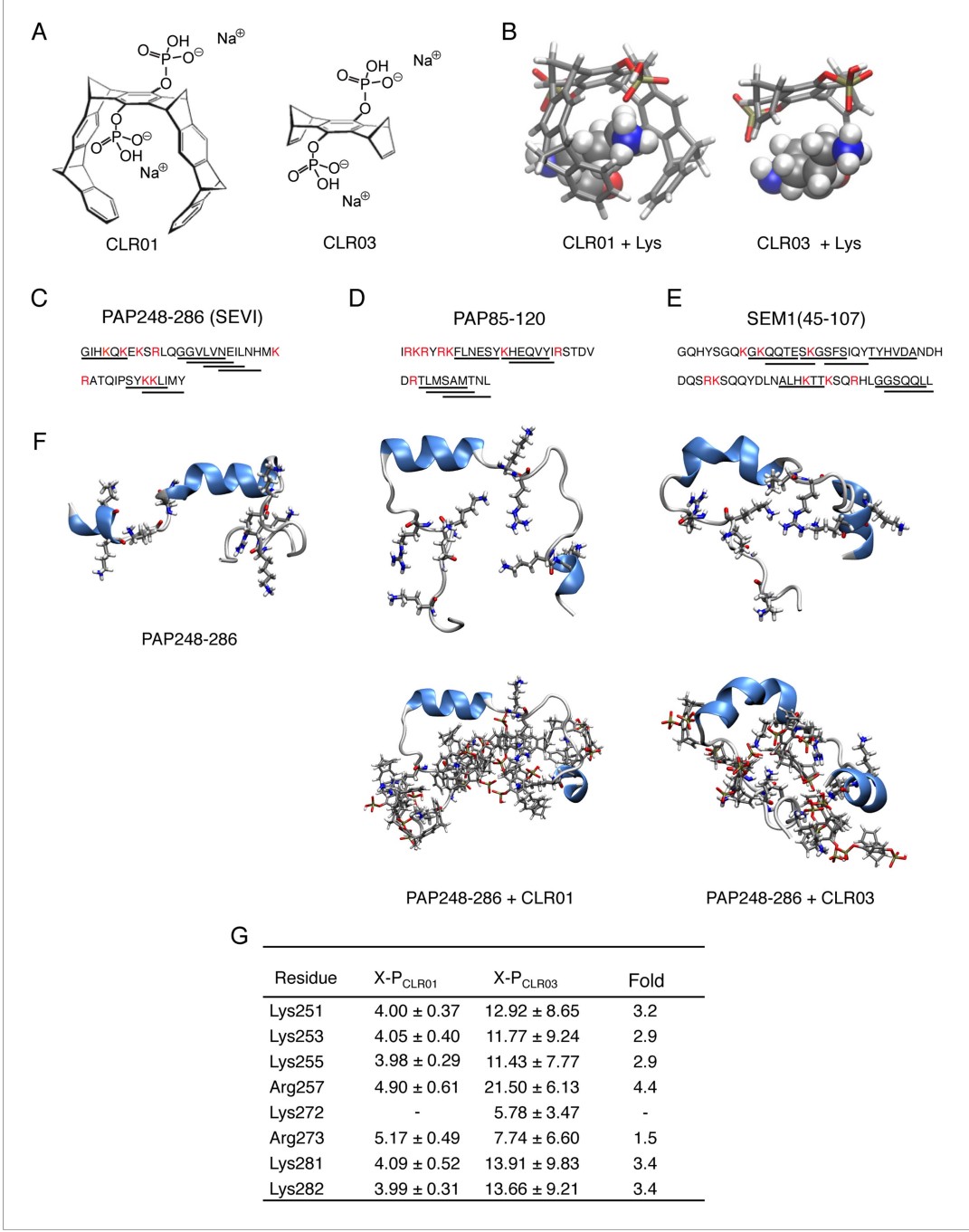

**Figure 1**. CLR01 binds to lysine and arginine residues. (**A**) Chemical structures of CLR01 and CLR03. (**B**) Stick representation of the structures of CLR01 and CLR03 and their engagement of lysine side chains. (**C**–**E**) The primary sequences of PAP248-286 (**C**), PAP85-120 (**D**), and SEM1(45-107) (**E**) are provided. Lysine and arginine residues are in red and hexapeptides predicted to form steric zippers (*Goldschmidt et al., 2010*; *Castellano and Shorter, 2012*) are underlined. (**F**) The average structures of the most populated clusters derived from the REMD simulations of PAP248-286 (left), PAP248-286 with 7 CLR01 molecules (middle), and PAP248-286 with 8 CLR03 molecules (right) are shown in the upper row, CLR01 and CLR03 molecules are not shown for clarity. The lower row shows, for each case, a representative structure of the most populated cluster including CLR01 and CLR03. (**G**) CLR03 establishes only labile interactions with PAP248-286 as shown by the large X-P distances (Å) between one P atom of CLR03 and the nitrogen atom of the lysine side chain (or carbon atom of the guanidinium moiety of arginine). Contrarily, the complexes between CLR01 and Lys or Arg were conserved during all the REMD simulations.

the cationic side chains into its torus-shaped cavity and engages the ammonium cation of lysine or the guanidinium cation of arginine with its anionic phosphate group in a tight ion pair (*Figure 1B*) (*Klärner and Schrader, 2013*). No other amino acids fulfill the requirements for this threading mechanism. The structure of the CLR01-lysine complex and the precise mechanism of lysine threading into the CLR01 guest cavity and subsequent ion pairing have been extensively characterized by NMR spectroscopy, crystal structure, molecular dynamics, and quantum mechanics/molecular mechanics (QM/MM) calculations (*Bier et al., 2013*; *Dutt et al., 2013*; *Klärner and Schrader, 2013*). Importantly, CLR01 appears only to complex with readily accessible lysine or arginine residues on protein surfaces, as evidenced by crystal structures and NMR experiments (*Bier et al., 2013*). This restriction makes CLR01 more selective for lysine or arginine residues found in intrinsically unfolded proteins or protein sequences.

Since amyloidogenic seminal peptides are particularly rich in lysine and arginine residues (*Roan et al., 2009*; *Arnold et al., 2012*; *Castellano and Shorter, 2012*) (*Figure 1C–E*, Lys and Arg residues are highlighted in red), we hypothesized that CLR01 might interfere with their HIV-enhancing activity. Here, we establish that CLR01 inhibits amyloidogenesis of PAP and SEM peptides, neutralizes the cationic surface charge of seminal amyloid, and rapidly remodels preformed SEVI and PAP85-120 fibrils. Strikingly, CLR01 also exhibits a direct antiviral effect by selectively disrupting the membrane of enveloped viruses. Thus, CLR01 represents an unprecedented candidate for further development as a microbicide as it not only inactivates HIV and other enveloped viruses but also antagonizes host-encoded seminal amyloids that enhance viral infection.

## Results

### CLR01 inhibits spontaneous assembly of seminal amyloid fibrils

Lysine residues in PAP248-286, PAP85-120, SEM1, and SEM2 peptides are frequently found within or immediately adjacent to hexapeptides predicted to form self-complementary β-strands (*Figure 1C–E*, underlined residues), termed steric zippers, which often comprise the spine of amyloid fibrils (*Nelson et al., 2005*; *Goldschmidt et al., 2010*; *Sievers et al., 2011*; *Castellano and Shorter, 2012*; *Frohm et al., 2015*). Moreover, the wealth of basic residues in PAP248-286, PAP85-120, and SEM1(45-107) (*Figure 1C–E*) led us to hypothesize that the lysine- and arginine-specific tweezer, CLR01, but not its derivative CLR03, which lacks hydrophobic sidewalls (*Sinha et al., 2011*) (*Figure 1A,B*), might bind to these residues and interfere with fibril assembly. To test this hypothesis, we first performed replica exchange molecular dynamics simulations using the available structure of PAP248-286, the best characterized of the amyloid-forming peptides in semen (*Münch et al., 2007*; *Castellano and Shorter, 2012*; *French and Makhatadze, 2012*). This analysis revealed that in silico, CLR01 bound at least seven of the eight positively charged residues in PAP248-286 without grossly altering peptide secondary structure (*Figure 1F*). Indeed, CLR01 engaged Lys251, Lys253, Lys281, and Lys282 (*Figure 1F,G*), which all reside in predicted steric zippers (*Castellano and Shorter, 2012*) (*Figure 1C*). Moreover, CLR01 bound Arg257, Lys281, and Lys282 (*Figure 1F,G*), which form part of the cross-β SEVI fibril core defined by hydrogen–deuterium exchange (*French and Makhatadze, 2012*). The CLR01 interaction was very similar among all the different lysine and arginine binding sites as indicated by similar (X-P) distances between the lysine ammonium or arginine guanidinium groups (X) and the bound phosphate group (P) in CLR01 (*Figure 1G*). By contrast, CLR03 only established more distant and variable interactions with lysine and arginine residues (*Figure 1G*). Thus, CLR01 but not CLR03 tightly shielded lysine and arginine residues in PAP248-286 suggesting that CLR01 could prevent PAP248-286 assembly into SEVI fibrils.

To test this prediction, CLR01 was assessed for its ability to inhibit the spontaneous amyloidogenesis (i.e., assembly in the absence of preformed fibril seeds) of PAP248-286, PAP85-120, and SEM1(45-107) (*Münch et al., 2007*; *Roan et al., 2011*; *Arnold et al., 2012*). Using the fluorescence of the diagnostic dye Thioflavin-T (ThT), which increases upon amyloid binding, we found that CLR01, but not CLR03, inhibited fibril assembly of all three peptides (*Figure 2A–C*). The half-maximal inhibitory concentrations (IC$_{50}$) of CLR01 inhibition of PAP248-286 and SEM1(45-107) fibrillization were ~1.19 μM and ~16.2 μM, respectively (*Figure 2D,F*). The IC$_{50}$ for inhibition of PAP248-286 fibrillization was significantly lower than the IC$_{50}$ for inhibition of SEM1(45-107) fibrillization (one-way ANOVA, p < 0.0001), suggesting that lysine, arginine, or both residues are more critical for PAP248-286 fibrillization. Moreover, inhibition of PAP248-286 and SEM1(45-107) fibrillization exhibited shallow dose–response curves with Hill slopes

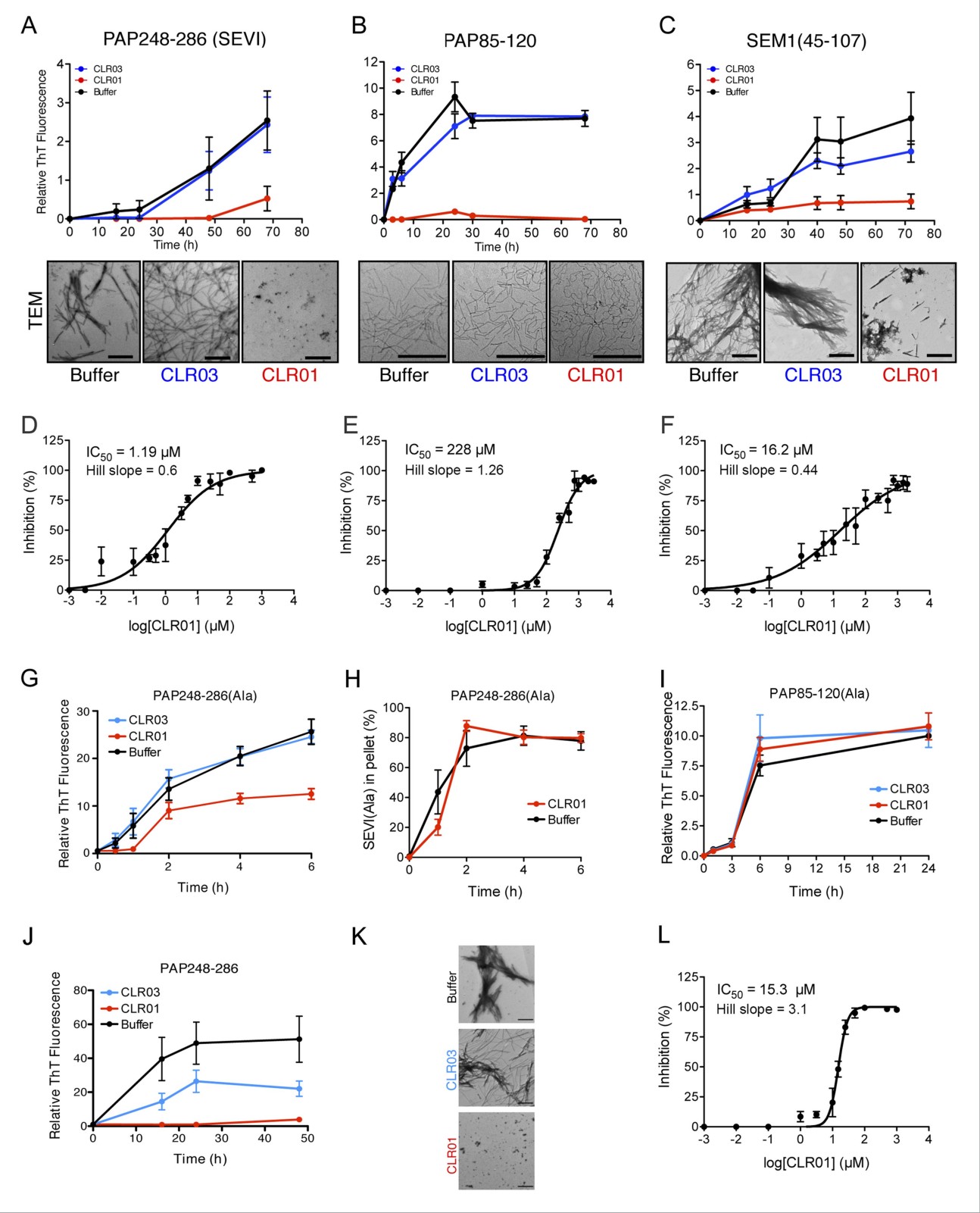

**Figure 2**. CLR01 inhibits formation of seminal amyloid fibrils. CLR01 inhibits fibril formation by PAP248-286 (1 mM) (**A**), PAP85-120 (1 mM) (**B**), and SEM1 (45-107) (500 μM) (**C**). Peptides were incubated with equimolar CLR01, CLR03, or buffer and agitated at 1400 rpm at 37°C. Aliquots were removed at various time points and fibrillization was assessed using the amyloid-binding dye, Thioflavin-T (ThT). Values represent means ±SEM (n = 4 for PAP248-286; n = 3 for PAP85-120; n = 9 for SEM1(45-107)). Transmission electron microscopy (TEM) images of PAP248-286 (**A**), PAP85-120 (**B**), and SEM1(45-107) (**C**) agitated

*Figure 2. continued on next page*

*Figure 2. Continued*

in the presence of CLR01, CLR03, or buffer. PAP248-286 and SEM1(45-107) samples were visualized after 72 hr of agitation, PAP85-120 after 24 hr. Scale bar: 500 nm. (**D**) Dose-response curve for CLR01 inhibition of PAP248-286 (1 mM) fibrillization after 72 hr of agitation. The $IC_{50}$ value and Hill slope are indicated. Values represent means ±SEM (n = 3–12). (**E**) Dose-response curve for CLR01 inhibition of PAP85-120 (1 mM) fibrillization after 24 hr of agitation. The $IC_{50}$ value and Hill slope are indicated. Values represent means ±SEM (n = 3–7). (**F**) Dose–response curve for CLR01 inhibition of SEM1(45-107) (500 µM) fibrillization after 72 hr of agitation. The $IC_{50}$ value and Hill slope are indicated. Values represent means ±SEM (n = 3–12). (**G, H**) CLR01 is unable to block PAP248-286(Ala) fibrillization. Lyophilized PAP248-286(Ala) was dissolved in PBS (100 µM), incubated with CLR01 (100 µM), CLR03 (100 µM), or buffer and agitated at 1400 rpm at 37˚C. Aliquots were removed at various time points and fibril assembly was assessed using ThT fluorescence (**G**) or sedimentation analysis (**H**). Values represent means ±SEM (n = 7–9) (**G**) or means ±SEM (n = 3) (**H**). (**I**) CLR01 is unable to block PAP85-120(Ala) fibrillization. Lyophilized PAP85-120(Ala) was dissolved in PBS (150 µM), incubated with CLR01 (150 µM), CLR03 (150 µM), or buffer and agitated at 1400 rpm at 37˚C. At the indicated times, fibril assembly was assessed using ThT fluorescence. Values represent means ±SEM (n = 3). (**J**) Inhibition of seeded PAP248-286 fibrillization by CLR01. Lyophilized PAP248-286 peptide was reconstituted at 1 mM in PBS and incubated with 1 mM CLR01, 1 mM CLR03, or buffer. Preformed SEVI fibrils (2% wt/wt) were added to each condition and agitated at 1400 rpm at 37˚C. Aliquots were removed at various time points and fibrillization was assessed using ThT fluorescence. Values represent means ±SEM (n = 6). (**K**) Electron microscopy visualization of CLR01 inhibition of seeded PAP248-286 fibrillization after 48 hr of agitation. Scale bar: 500 nm. (**L**) Dose-response curve for CLR01 inhibition of PAP248-286 fibrillization seeded by preformed SEVI fibrils (2% wt/wt) after 48 hr of agitation. The $IC_{50}$ value and Hill slope are indicated. Values represent means ±SEM (n = 3–7).

The following figure supplements are available for figure 2:

**Figure supplement 1**. CLR01 does not displace ThT from SEVI, PAP85-120, or SEM1(45-107) fibrils.

**Figure supplement 2**. CLR01 does not impede adsorption of SEVI, PAP85-120, or SEM1(45-107) fibrils to the EM grid.

**Figure supplement 3**. Lysine and poly-L-lysine antagonize the ability of CLR01 to inhibit spontaneous formation of seminal amyloid fibrils.

**Figure supplement 4**. BSA or preclearing CLR01 has no effect on the ability of CLR01 to inhibit formation of seminal amyloid fibrils.

**Figure supplement 5**. Lysine or poly-L-lysine, but not BSA or preclearing CLR01 solutions, antagonize the ability of CLR01 to inhibit seeded assembly of SEVI fibrils.

of ~0.6 for PAP248-286 and ~0.44 for SEM1(45-107) (*Figure 2D,F*), indicating negative co-operativity or CLR01 binding to multiple sites with different affinities. Importantly, these shallow dose responses provide evidence against a non-specific mechanism of inhibition involving colloidal CLR01 aggregates, as aggregating inhibitors tend to exhibit steep dose response curves (*Shoichet, 2006*). The $IC_{50}$ of CLR01 to impede PAP85-120 assembly was significantly higher at ~228 µM (one-way ANOVA, p < 0.0001; *Figure 2E*). The higher $IC_{50}$ is likely due to the lower number of lysine or arginine residues in the PAP85-120 sequence located in hexapeptides predicted to have high amyloid propensity (*Figure 1D*) (*Castellano and Shorter, 2012*). Indeed, PAP85-120 has 1 lysine or arginine located in predicted steric zippers, whereas PAP248-286 has 4 and SEM1(45-107) has 3 (*Figure 1D*). The Hill slope was ~1.26, indicating weak positive co-operativity (*Figure 2E*), but was still below the ~1.5–2 range that might indicate a mechanism of inhibition involving colloidal CLR01 aggregates (*Shoichet, 2006*).

We examined the possibility that CLR01 might simply displace ThT from fibrils by employing a ThT displacement assay (*Lockhart et al., 2005*). Thus, preformed SEVI, PAP85-120, or SEM1(45-107) fibrils were preincubated with ThT. Buffer, or an excess of CLR01 or a known competitor of ThT binding, BTA-1, were then added and ThT displacement was assessed by fluorescence measurements (*Lockhart et al., 2005*). ThT fluorescence decreased drastically in the presence of BTA-1, confirming its ability to displace ThT from fibrils (*Lockhart et al., 2005*), whereas CLR01 and buffer had no effect (*Figure 2—figure supplement 1*). These findings suggest that CLR01 does not simply displace ThT from SEVI, PAP85-120, or SEM1(45-107) fibrils. Thus, any reduction in ThT fluorescence caused by CLR01 can be attributed to an inhibition of fibril assembly.

Transmission electron microscopy (TEM) analysis confirmed that CLR01 prevented assembly of PAP248-286 and SEM1(45-107) into mature fibrils (*Figure 2A,C*). Only small oligomeric forms (*Figure 2A*) or amorphous aggregates in combination with sparse short fibrils (*Figure 2C*) were detectable in the presence of CLR01, as opposed to abundant fibrils observed in the presence of buffer or CLR03 (*Figure 2A,C*). The effect of CLR01 on PAP85-120 assembly was less apparent by

TEM (*Figure 2B*). However, the PAP85-120 assemblies formed in the presence of CLR01 appeared more flexible and curvilinear, and differed from the rigid, straight fibrils formed in the presence of CLR03 or buffer (*Figure 2B*). Since the structures that formed in the presence of CLR01 were not ThT-reactive (*Figure 2B*) they most likely represent non-amyloid aggregates. Thus, CLR01 appears to impede the transition of PAP85-120 to mature amyloid fibrils and abrogates formation of SEVI and SEM1(45-107) fibrils.

We excluded the possibility that CLR01 might impede adsorption of SEVI, PAP85-120, or SEM(45-107) fibrils to the EM grid. When we mixed CLR01 (or buffer) with preformed SEVI, PAP85-120, or SEM1(45-107) fibrils for 10 min (a time at which no fibril remodeling occurs; *Figure 3A–C*) and then adsorbed them to the grid, we observed abundant fibrils in both the CLR01 and buffer control conditions (*Figure 2—figure supplement 2*). Thus, any reduction in the presence of fibrils observed by EM can be attributed to inhibition of fibril assembly.

To investigate the role of lysine and arginine interactions with CLR01 in the inhibition of fibril assembly, we examined PAP248-286(Ala) or PAP85-120(Ala), two peptide analogues in which all lysine and arginine residues are replaced by alanine (*Roan et al., 2009*). PAP248-286(Ala) formed amyloid fibrils in the presence of CLR01, CLR03, and buffer (*Figure 2G*). To further analyze the effect of CLR01 on PAP248-286(Ala) assembly, we employed a sedimentation assay. This assay revealed that equal amounts of PAP248-286(Ala) entered the pellet fraction when the peptide was incubated with buffer or CLR01, indicating that the formation of PAP248-286(Ala) amyloid fibrils is unaffected by CLR01 treatment (*Figure 2H*). Thus, it is likely that in the presence of CLR01, PAP248-286(Ala) assembles into a subtly distinct set of cross-beta structures or fibril 'strains' that are less ThT-reactive (*Figure 2G,H*). Amyloidogenesis by PAP85-120(Ala) was also unaffected by CLR01 (*Figure 2I*). The PAP248-286(Ala) and PAP85-120(Ala) peptides spontaneously assemble into amyloid more rapidly than the wild-type peptides (*Figure 2A,B,G,I*). However, we have been unable to establish conditions (e.g., higher CLR01 concentrations) where CLR01 prevents assembly of PAP248-286(Ala) and PAP85-120(Ala) into amyloid fibrils (data not shown). These findings suggest that CLR01-lysine contacts, CLR01-arginine contacts, or both, are essential for inhibition of fibrillization.

Next, we tested if an excess of free lysine or poly-L-lysine could interfere with the ability of CLR01 to block fibrillization of PAP248-286, PAP85-120, and SEM1(45-107) (*Figure 2—figure supplement 3*). Both 20 mM lysine and 1 mM poly-L-lysine completely abrogated CLR01 inhibition of fibril formation by these three peptides. These results strongly support direct interaction of the tweezer with lysine, arginine, or both residues in PAP248-286, PAP85-120, and SEM1(45-107) as the underlying mechanism of CLR01 inhibition. We suggest that these specific interactions preclude the conformational rearrangements necessary for amyloidogenesis.

Some small molecules must form higher order colloidal aggregates to inhibit amyloid assembly (*Feng et al., 2008*; *Young et al., 2015*). BSA inhibits the activity of colloidal small-molecule aggregates via adsorption (*McGovern et al., 2002*; *Coan and Shoichet, 2007*; *Feng et al., 2008*). Additionally, colloidal small-molecule aggregates can be precleared via centrifugation (*McGovern et al., 2003*). Thus, we used these two techniques to test whether inhibition of PAP248-286, PAP85-120, or SEM1(45-107) aggregation could be mediated via unexpected colloid formation by CLR01. CLR01 was found to inhibit PAP248-286, PAP85-120, and SEM1(45-107) fibril assembly in the presence of BSA (10 mg/ml) or when solutions containing CLR01 were centrifuged at 16,100×$g$ for 20 min before adding the supernatant solution to the proteins (*Figure 2—figure supplement 4*). Thus, CLR01 is not inhibiting PAP248-286, PAP85-120, and SEM1(45-107) fibril assembly via a mechanism that involves colloidal CLR01 aggregates.

## CLR01 inhibits assembly of SEVI seeded by preformed fibrils

The addition of a small amount of preformed SEVI fibrils to soluble PAP248-286 seeds polymerization and eliminates the lag phase for assembly (*Ye et al., 2009*) (compare *Figure 2A* with *Figure 2J*). Remarkably, in addition to obstructing unseeded PAP248-286 assembly (*Figure 2A*), CLR01 also completely inhibited seeded fibrillization (*Figure 2J,K*). Dose–response analysis established an IC$_{50}$ of ~15.3 μM for CLR01 inhibition of seeded PAP248-286 assembly (*Figure 2L*) compared to ~1.19 μM for spontaneous PAP248-286 assembly (*Figure 2D*). Thus, higher concentrations of CLR01 are required to inhibit PAP248-286 assembly once SEVI fibrils have formed. Interestingly, the Hill slope for inhibition of seeded PAP248-286 assembly was ~3.1 indicating

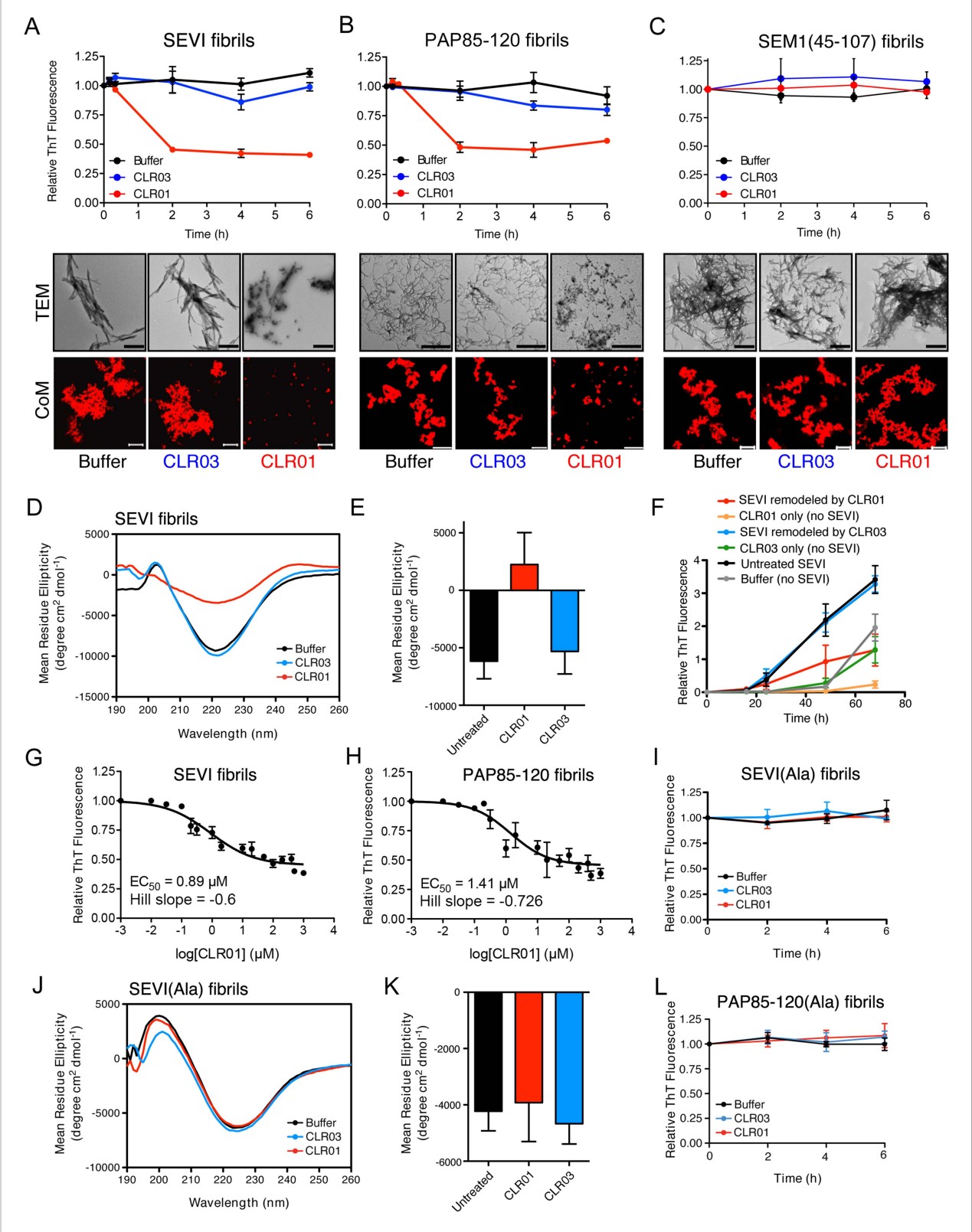

**Figure 3**. CLR01 rapidly remodels SEVI and PAP85-120 fibrils. (**A–C**) Preformed SEVI fibrils (**A**), PAP85-120 fibrils (**B**), and SEM1(45-107) fibrils (**C**) (20 µM) were treated with a 10-fold excess of CLR01 or CLR03 or buffer for 0–6 hr at 37°C. Fibril integrity was assessed using ThT. Values represent means ±SEM (n = 3). TEM (middle panel) and confocal microscopy (bottom panel) of SEVI (**A**), PAP85-120 fibrils (**B**), and SEM1(45-107) fibrils (**C**) obtained after 2 hr treatment with CLR01,

*Figure 3. continued on next page*

*Figure 3. Continued*

CLR03, or buffer. Scale bar for TEM images: 500 nm. For confocal microcopy (CoM), samples were stained with Proteostat dye. Scale bar: 20 μm. (**D**) CD spectrum of 50 μM SEVI fibrils incubated with 50 μM CLR01, 50 μM CLR03, or buffer. A representative spectrum is shown. (**E**) The mean residue ellipticity (MRE) at 218 nm was averaged from three independent experiments. Values represent means ±SEM. (**F**) Seeding with CLR01-remodeled SEVI products. SEVI fibrils (20 μM) were treated with 200 μM CLR01, 200 μM CLR03, or buffer for 3 hr and reaction products were used to seed PAP248-286 fibrillization (0.1% fibril seed, 1 mM peptide). Buffer conditions with no fibril seed present were also included. Fibrillization was monitored by ThT fluorescence. Values represent means ±SEM (n = 8). (**G**) Dose–response curve for CLR01 remodeling of SEVI fibrils (20 μM) after 2 hr of treatment. The $EC_{50}$ value and Hill slope are indicated. Values represent means ±SEM (n = 3–15). (**H**) Dose-response curve for CLR01 remodeling of PAP85-120 fibrils (20 μM) after 2 hr of treatment. The $EC_{50}$ value and Hill slope are indicated. Values represent means ±SEM (n = 3–14). (**I**) CLR01 is unable to remodel SEVI(Ala) fibrils. Preformed SEVI(Ala) fibrils (20 μM) were treated with 200 μM CLR01, 200 μM CLR03, or buffer for 0–6 hr. Fibril integrity was assessed using ThT fluorescence. Values represent means ±SEM (n = 3). (**J**) CD spectrum of 50 μM SEVI(Ala) fibrils incubated with either 50 μM CLR01, 50 μM CLR03, or buffer. One representative example is shown. (**K**) The mean residue ellipticity (MRE) at 218 nm was averaged from three independent experiments. Values represent means ±SEM (n = 3). (**L**) Preformed PAP85-120(Ala) fibrils (20 μM) were treated with CLR01 or CLR03 (200 μM) or buffer for 0–6 hr. Fibril integrity was assessed using ThT fluorescence. Values represent means ±SEM (n = 3).

The following figure supplements are available for figure 3:

**Figure supplement 1**. Lysine and poly-L-lysine antagonize the ability of CLR01 to remodel SEVI and PAP85-120 fibrils.

**Figure supplement 2**. BSA or preclearing CLR01 has no effect on the ability of CLR01 to remodel SEVI and PAP85-120 fibrils.

positive co-operativity and was steeper than the Hill slope of ∼0.6 for inhibition of spontaneous PAP248-286 fibrillization (*Figure 2D,L*). This dissimilarity might indicate a mechanistic difference in how CLR01 inhibits seeded and spontaneous PAP248-286 fibrillization. However, this difference is not likely to be due colloidal CLR01 aggregates, as inhibition of seeded assembly by CLR01 was unaffected by BSA or by preclearing CLR01 solutions via centrifugation (*Figure 2—figure supplement 5*). Rather, we suggest that the CLR01-mediated inhibition of fibril elongation in seeded SEVI assembly has requirements distinct from the CLR01-mediated inhibition of initial fibril nucleation in spontaneous, unseeded SEVI assembly.

Inhibition of seeded PAP248-286 assembly was alleviated by excess lysine or poly-L-lysine (*Figure 2—figure supplement 5*). Thus, inhibition of seeded PAP248-286 assembly by CLR01 likely requires specific interaction with lysine, arginine, or both residues. Importantly, because CLR01 impedes both unseeded and seeded PAP248-286 assembly, it likely acts at multiple stages of the fibrillization process including the initial nucleation and fibril elongation steps. These observations also indicate that lysine, arginine, or both residues play an important role in the primary nucleation and subsequent elongation of PAP248–286 fibrils.

## CLR01 remodels preformed SEVI and PAP85-120 fibrils

Semen-derived fibrils are abundant in liquefied fresh ejaculates (*Usmani et al., 2014*). Thus, agents that not only inhibit fibril formation but also remodel preformed fibrils would be advantageous for microbicide development (*Castellano and Shorter, 2012*). To test whether CLR01 could remodel seminal amyloid, SEVI, PAP85-120 and SEM1(45-107) fibrils were treated with CLR01 or CLR03, and ThT fluorescence intensity was monitored. Brief incubations with CLR01 (10–20 min) had no effect, indicating that CLR01 did not simply displace ThT from fibrils (*Figure 2—figure supplement 1*; *Figure 2—figure supplement 2*; *Figure 3A–C*). After 2 hr, however, CLR01 treatment of SEVI (*Figure 3A*) and PAP85-120 (*Figure 3B*) but not SEM1(45-107) (*Figure 3C*) fibrils resulted in a reduction in ThT fluorescence intensity by more than 50%. Even after 24 hr, SEM1(45-107) fibrils were not remodeled by CLR01 (data not shown). SEVI and PAP85-120 fibril remodeling was confirmed by TEM, which showed few intact fibrils and predominately smaller nonfibrillar species after CLR01 treatment (*Figure 3A,B*). By contrast, TEM revealed that SEM1(45-107) fibrils were still abundant after prolonged CLR01 treatment (*Figure 3C*). This observation confirms that CLR01 does not simply prevent fibrils from adsorbing to the EM grid (*Figure 2—figure supplement 2*). Similar results were obtained by fluorescence microscopy (CoM) of samples stained with Proteostat, a red fluorescent aggregate sensing dye (*Figure 3A–C*) (*Usmani et al., 2014*). These effects of CLR01 were remarkably rapid in comparison to the slow disassembly of Aβ or α-synuclein

fibrils by CLR01, which required several weeks (*Sinha et al., 2011*; *Prabhudesai et al., 2012*). Thus, CLR01 can remodel SEVI and PAP85-120 fibrils on a time scale that would be useful for prevention of HIV infection, which usually takes place within the first hours after deposition of semen in the anogenital tract (*Shattock and Moore, 2003*).

We next performed circular dichroism experiments to examine the effect of CLR01 on the amyloid cross-β structure. Untreated SEVI fibrils or SEVI fibrils treated with CLR03 exhibited a pronounced minimum indicative of a characteristic β-sheet rich structure (*Figure 3D,E*). By contrast, SEVI fibrils incubated with CLR01 showed a loss in the β-sheet minimum (*Figure 3D,E*), confirming that CLR01 alters the cross-β architecture of SEVI fibrils. Furthermore, SEVI fibrils remodeled by CLR01 were less effective seeds for polymerization of monomeric PAP248-286 (*Figure 3F*). Thus, CLR01 remodels SEVI fibrils into alternative non-templating conformers.

A dose–response curve for CLR01-mediated remodeling of SEVI and PAP85-120 fibrils revealed a half-maximal effective concentration ($EC_{50}$) value of ~0.89 μM and ~1.41 μM, respectively (*Figure 3G,H*). Interestingly, in both cases the Hill slope was similar, approximately −0.6 for SEVI and approimately −0.726 for PAP85-120, indicating negative co-operativity or CLR01 binding to multiple sites with different affinities. These shallow dose–response curves provide evidence against a colloidal mechanism of CLR01 action (*Shoichet, 2006*). Previous studies have reported that a 10-fold molar excess of CLR01 could remodel preformed Aβ40, Aβ42, or α-synuclein fibrils slowly over a time period of weeks (*Sinha et al., 2011*; *Prabhudesai et al., 2012*). It is therefore remarkable that CLR01 remodels SEVI and PAP85-120 fibrils with such alacrity and at sub-stoichiometric concentrations (relative to peptide monomers).

To elucidate the role of lysine- and arginine-tweezer interactions in CLR01-mediated remodeling of SEVI and PAP85-120 fibrils, experiments were performed with fibrils comprised of PAP248-286(Ala) or PAP85-120(Ala). CLR01 was unable to remodel preformed SEVI(Ala) or PAP85-120(Ala) fibrils (*Figure 3I–L*). Indeed, we have been unable to establish conditions (e.g., higher CLR01 concentrations) where CLR01 remodels SEVI(Ala) or PAP85-120(Ala) fibrils (data not shown). Furthermore, addition of excess free lysine or poly-L-lysine to CLR01 prevented its SEVI and PAP85-120 amyloid-remodeling activity (*Figure 3—figure supplement 1*). Thus, CLR01-lysine, CLR01-arginine, or both contacts are critical for the remodeling process. By contrast, CLR01 retained SEVI and PAP85-120 amyloid-remodeling activity in the presence of BSA (10 mg/ml) or if CLR01 solutions were precleared via centrifugation (*Figure 3—figure supplement 2*). Thus, colloidal CLR01 aggregates do not contribute to the amyloid-remodeling activity of CLR01. Taken together, our findings suggest that the lysine- and arginine-specific molecular tweezer CLR01 prevents formation of seminal amyloid and remodels mature PAP85-120 and SEVI fibrils via binding to lysine and arginine residues.

## CLR01 abrogates the interaction of seminal amyloids with viral particles and diminishes their infection enhancing property

Since the infectivity-enhancing activity of seminal amyloid fibrils is due to their positive surface charge (*Roan et al., 2009*; *Arnold et al., 2012*), we examined next whether CLR01 affected this property. To prevent fibril remodeling from occurring in these experiments, fibrils were mixed with CLR01, CLR03, or buffer, and samples were immediately centrifuged to remove unbound CLR01 and CLR03 molecules. Subsequently, the surface charge of resuspended fibrils was determined by zeta potential measurements. We found that CLR01, but not CLR03, neutralized the positive surface charge of SEVI, PAP85-120, and SEM1(49-107) fibrils within minutes (*Figure 4A*). Notably, pre-treatment of CLR01 with lysine or poly-L-lysine abrogated the fibril neutralizing activity of CLR01 (*Figure 4—figure supplement 1*), which further supports previous findings concerning the specificity underlying these interactions (*Fokkens et al., 2005*; *Klärner et al., 2006, 2010*). Next, confocal microscopy was used to assess whether this neutralization could abrogate fibril binding to YFP-tagged virions. As previously shown (*Roan et al., 2011*; *Arnold et al., 2012*; *Yolamanova et al., 2013*), buffer-treated fibrils efficiently sequestered virions (*Figure 4B*). In contrast, fibrils pretreated with CLR01, but not CLR03, were unable to form fibril–virion complexes (*Figure 4B*).

Since fibril–virion complexes are already present before ejaculation or form rapidly post emission (*Usmani et al., 2014*), we next investigated whether CLR01 could displace virions from preformed fibril–virion complexes. Remarkably, CLR01 but not CLR03 substantially reduced the number of virions covering the surface of SEVI, PAP85-120, and SEM1(49-107) fibrils (*Figure 4C*). We excluded the

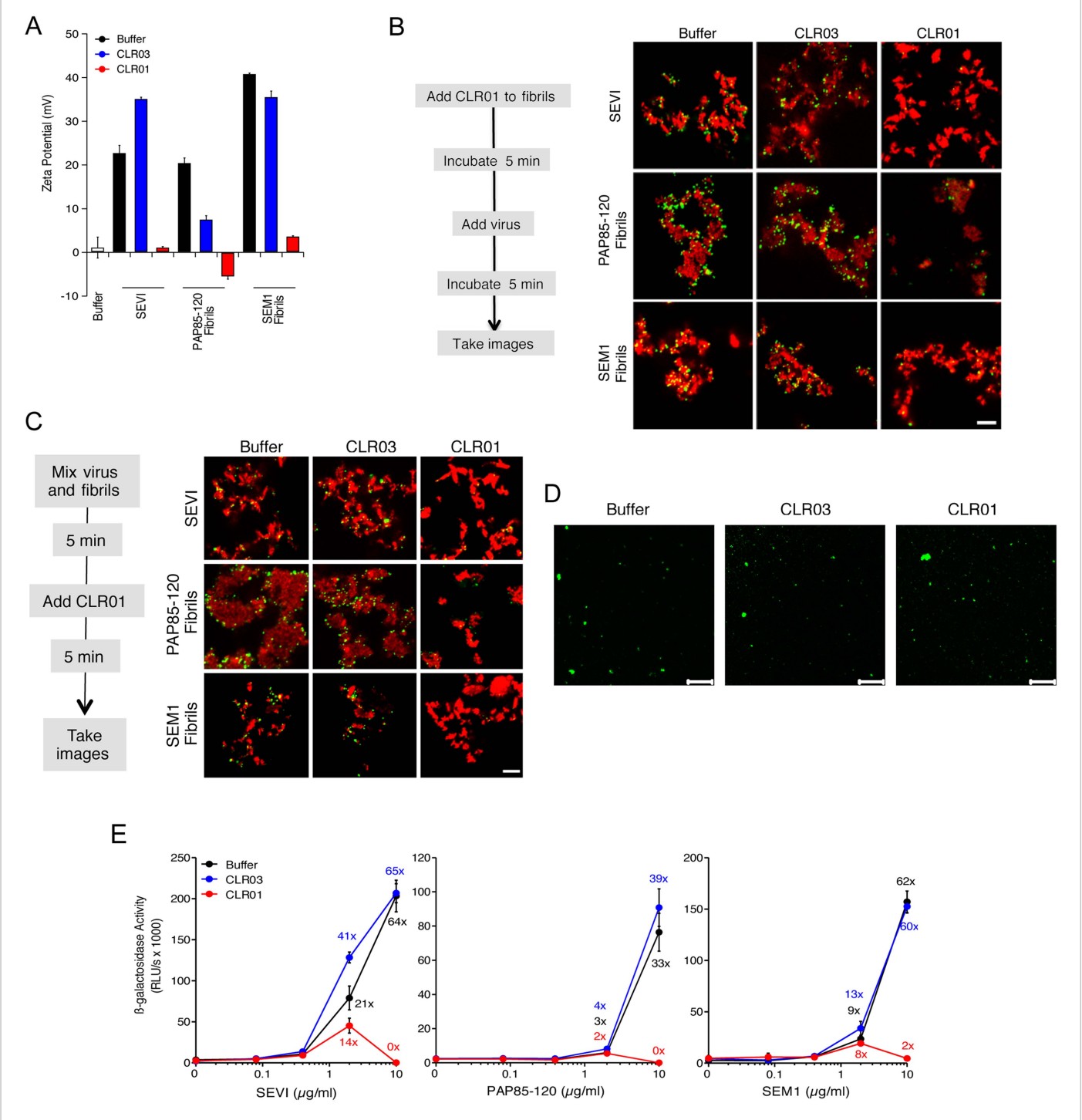

**Figure 4**. CLR01 neutralizes the positive surface charge of seminal amyloids and abrogates their ability to bind virions and enhance HIV infection. (**A**) Surface charge of seminal amyloids determined by zeta potential measurements. Fibrils were mixed with buffer, CLR01, or CLR03 and centrifuged for 10 min at 20,000×g. The pellets were resuspended in 1 mM KCl and zeta potential was measured. Values represent means ±SD (n = 3). (**B**) Fibrils (200 µg/ml) were pretreated with PBS, CLR01, or CLR03 in 20-fold excess for 5 min and stained with Proteostat dye. MLV-Gag-YFP particles (green) were added 1:2 and allowed to incubate with pretreated fibrils (red) for 5 min. Samples were analyzed using confocal microscopy. Scale bar: 5 µm. (**C**) CLR01 displaces virions from fibrils. MLV-Gag-YFP particles (green) were incubated with Proteostat-stained seminal amyloids (red) for 5 min. PBS, CLR01, or CLR03 were added at 20-fold excess, and after an additional 5 min of incubation, samples were analyzed by confocal microscopy. Scale bar: 5 µm. (**D**) CLR01 does not quench the fluorescence signal of MLV-Gag-YFP particles. MLV-Gag-YFP particles were treated with 300 µM CLR01 or CLR03 for

*Figure 4. continued on next page*

*Figure 4. Continued*

30 min at 37°C. Thereafter, virions were recovered in the pellet fraction via high-speed centrifugation and aliquots analyzed by confocal microscopy. Scale bar = 20 µm. (**E**) CLR01 antagonizes the HIV-1 enhancing activity of seminal amyloids. SEVI, PAP85-120, and SEM1(49-107) fibrils were mixed with a 20-fold excess of CLR01 or CLR03. CCR5-tropic HIV-1 was added and samples were used to inoculate TZM-bl cells. Infection rates were determined 3 days post infection. Values represent mean β-galactosidase activities derived from triplicate infections ±SD (RLU/s: relative light units per second). Numbers above symbols indicate the n-fold enhancement of infection.

The following figure supplements are available for figure 4:

**Figure supplement 1**. Lysine and poly-L-lysine dose-dependently antagonize CLR01 binding to SEVI.

**Figure supplement 2**. CLR01 does not quench the fluorescence signal of MLV-Gag-YFP particles.

**Figure supplement 3**. CLR01 is not cytotoxic.

possibility that CLR01 quenches the fluorescence of YFP-tagged virions by analyzing virions treated with CLR01 or controls by confocal microscopy. We found that treatment of virions with CLR01 did not affect virion fluorescence (*Figure 4D*) or the number of fluorescent particles (*Figure 4—figure supplement 2*). However, CLR01-treated virions were dispersed throughout the chamber indicating that their biophysical properties might be altered (*Figure 4—figure supplement 2*). Our results demonstrate that CLR01 not only prevents the interaction of fibrils with virions but also displaces viral particles from preformed fibril–virion complexes.

The combined effects of CLR01 on fibril architecture (*Figures 2, 3*) and the formation of fibril–virion complexes (*Figure 4A–C*) led us to investigate whether the tweezer might diminish the infection-enhancing property of seminal amyloids. First, we analyzed possible cytotoxic effects of the tweezer in TZM-bl cells, a HIV reporter cell line commonly used to study virus infection (*Münch et al., 2007*; *Usmani et al., 2014*). We found that CLR01 and CLR03 did not cause cytotoxic effects at concentrations up to 500 µM, whereas the surfactants Triton X-100, nonoxynol 9 and sodium dodecyl sulfate (SDS) were highly toxic (*Figure 4—figure supplement 3*). Thus, CLR01 does not act in a manner similar to non-ionic or anionic surfactants.

Next, preformed fibrils were treated for 5 min with CLR01, CLR03, or buffer. After addition of a low dose of CCR5-tropic HIV, the resulting mixture was added to TZM-bl cells and infection was measured via β-galactosidase activity 3 days later. As previously reported (*Münch et al., 2007*; *Roan et al., 2011*; *Arnold et al., 2012*), the three semen-derived amyloids augmented HIV infection in a dose-dependent manner with maximal enhancements between 33- to 64-fold (*Figure 4E*). Remarkably, pretreatment of SEVI, PAP85-120, and SEM1(49-107) amyloid with CLR01 eliminated the infection-enhancing property of the fibrils, while CLR03 had no effect (*Figure 4E*). Thus, CLR01 abrogates the infection enhancing activity of seminal amyloids.

## CLR01 exhibits direct anti-HIV activity

In addition to engaging amyloid fibrils (*Figures 3, 4A*), CLR01 also appeared to have a direct effect on viral particles (*Figure 4—figure supplement 2*). To study the consequences of this interaction in more detail, a CXCR4- and CCR5-tropic HIV-1 NL4-3 recombinant (*Papkalla et al., 2002*) and two CCR5-tropic transmitted/founder viruses (*Ochsenbauer et al., 2012*) were pretreated with CLR01 or controls and then examined for infectivity. Remarkably, CLR01 but not CLR03 abrogated viral infectivity in a dose-dependent manner with $IC_{50}$ values ranging from ~13.7 to 20.1 µM (*Figure 5A*). This reduction in infectivity was only observed when virions were exposed to CLR01 and not when target cells were pretreated with the tweezer (*Figure 5B*). Thus, in addition to its anti-amyloid properties, CLR01 also has direct anti-HIV activity that is independent of the viral co-receptor tropism or strain.

Exposure of CLR01 to poly-L-lysine prior to incubation with virions abrogated its anti-HIV-1 activity in a dose-dependent manner (*Figure 5—figure supplement 1*). Near-complete inhibition was achieved with 75 nM poly-L-lysine. Thus, the lysine-binding cavity of CLR01 plays an important role in the antiviral (*Figure 5*; *Figure 5—figure supplement 1*) and anti-amyloid (*Figure 2*; *Figure 2—figure supplements 3, 5*; *Figure 3*; *Figure 3—figure supplement 1*) activities of CLR01.

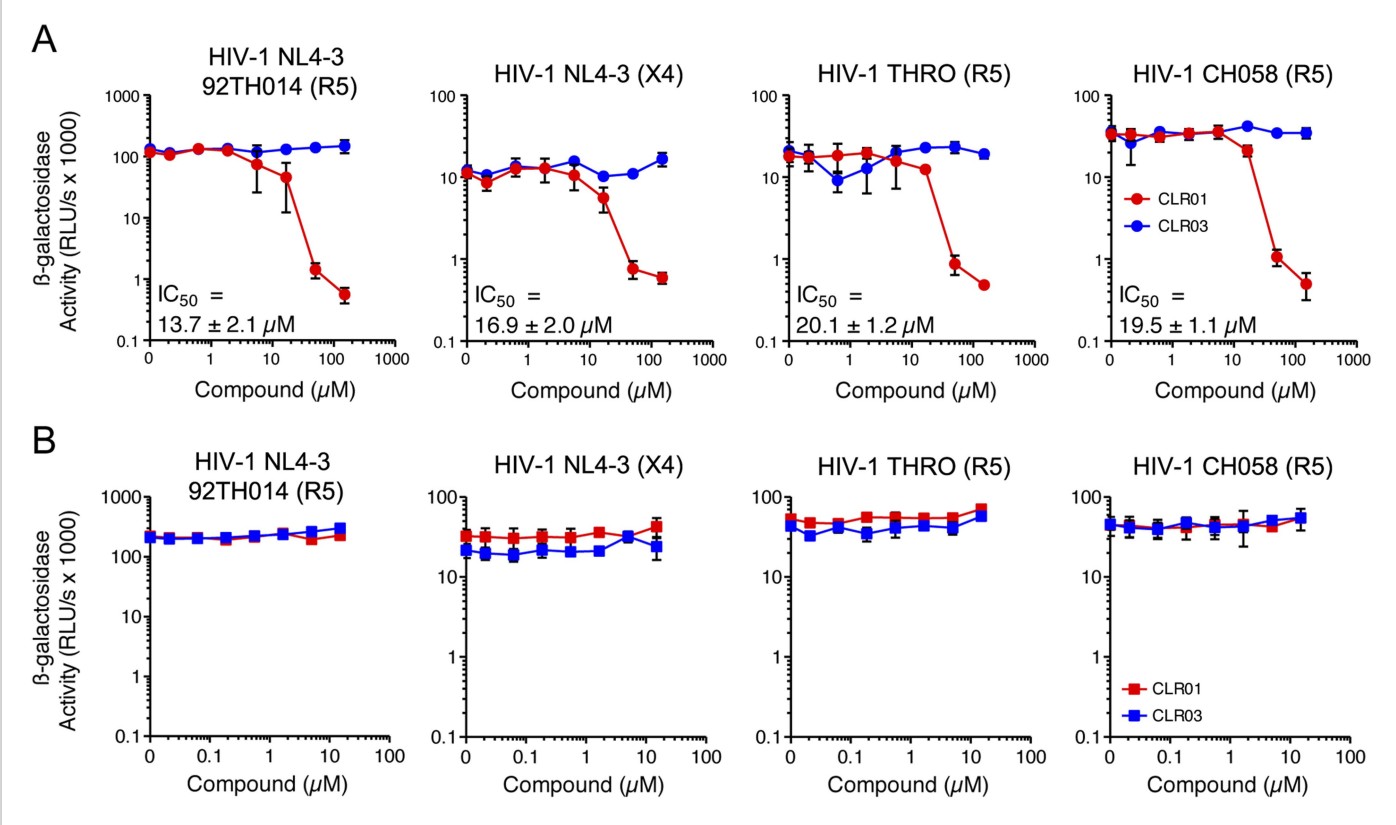

**Figure 5**. CLR01 has direct anti-HIV-1 activity. (**A**) CLR01 blocks HIV-1 infection by targeting virions. The CXCR4 (X4)-tropic lab-adapted HIV-1 NL4-3 strain, a CCR5 (R5)-tropic V3 loop recombinant thereof, and two R5-tropic transmitted/founder viruses (THRO and CH058) were incubated with CLR01 or CLR03 for 10 min and then used to infect TZM-bl cells. Infection rates were determined 3 days post infection. Values represent mean β-galactosidase activities derived from triplicate infections ±SD (RLU/s: relative light units per second). (**B**) The antiviral activity of CLR01 is not directed against the cell. TZM-bl cells were exposed to CLR01 or CLR03 for 1 hr. Thereafter, cell culture medium was removed and cells were infected with the indicated viruses. Infection rates were determined 3 day post infection. Values represent mean β-galactosidase activities derived from triplicate infections ±SD (RLU/s: relative light units per second).

The following figure supplements are available for figure 5:

**Figure supplement 1**. Poly-L-lysine counteracts the antiviral activity of CLR01.

**Figure supplement 2**. Coated BSA does not antagonize the antiviral activity of CLR01.

**Figure supplement 3**. BSA does not antagonize the antiviral activity of CLR01.

**Figure supplement 4**. Preclearing CLR01 via centrifugation does not affect antiviral activity.

Incubation of the tweezer in solutions containing up to 5% (sevenfold molar excess) coated or free BSA did not reduce the anti-HIV activity of CLR01 (*Figure 5—figure supplements 2, 3*). Similarly, the antiviral activity of CLR01 was not reduced when CLR01 solutions were precleared of any aggregates via centrifugation (*Figure 5—figure supplement 4*). Moreover, the pellet fraction did not possess antiviral activity (*Figure 5—figure supplement 4*). These data strongly suggest that colloidal CLR01 aggregates do not contribute to the direct antiviral activity of CLR01.

## Antiviral mechanism of CLR01

To define the underlying mechanism of this antiviral activity, we tested whether CLR01 disrupts the integrity of the viral membrane, leading to the release of the inner viral p24 capsid protein. HIV virions

were exposed to buffer, CLR01 or CLR03 and then separated by centrifugation into a soluble fraction (containing free p24) and a sedimentable fraction (containing intact viral particles). ELISA measurements demonstrated that the amount of p24 was increased in the soluble fraction of CLR01-treated samples as compared to samples treated with CLR03 or buffer (*Figure 6A*). Time course experiments revealed that a 5-min incubation of virus with 10 μM CLR01 resulted in a 62% decrease in HIV infectivity, and a 10-min incubation achieved almost a 100% reduction (*Figure 6B*). Atomic force microscopy (AFM) of mouse leukemia virus particles (MLV) confirmed that treatment with CLR01 destroyed virion architecture (*Figure 6C,D*). This effect was independent of the presence of viral glycoproteins, since CLR01 also destroyed HIV-1 particles lacking gp120/41 (Δenv) (*Figure 6C*), suggesting that CLR01 disrupts the integrity of the viral membrane.

The result that CLR01 destroys retroviral particles with $IC_{50}$ values between ~10 and 20 μM by compromising virion integrity was unexpected, particularly because the tweezer does not affect cell viability at these concentrations (*Figure 4—figure supplement 3*) (*Sinha et al., 2011*; *Attar et al., 2012*; *Herzog et al., 2015*). Viral membranes differ from cellular membranes in that they are two to threefold enriched in specific lipids, such as sphingomyelin and cholesterol, which can form microdomains termed lipid rafts (*Aloia et al., 1988*; *Brügger et al., 2006*; *Chan et al., 2008*; *Lorizate et al., 2009*, *2013*; *Gerl et al., 2012*). We hypothesized that CLR01 might selectively disrupt lipid-raft enriched membranes. To test this hypothesis, dye-loaded giant unilamellar vesicles (GUV) were generated that consisted of either 1,2-dioleoyl-*sn*-glycero-3-phosphocholine (DOPC) to recapitulate a bilayer devoid of lipid rafts or a mixture of DOPC, sphingomyelin and cholesterol (DOPC/SM/Chol) to more closely resemble a lipid raft-rich viral membrane. Strikingly, CLR01 permeabilized the model viral membrane within 5–10 min while exhibiting no effect on the DOPC membrane, even after 60 min of incubation (*Figure 6E*). These results were confirmed by AFM at higher spatial resolution (*Figure 6F*). CLR01 treatment of a DOPC membrane that was deposited on mica surface did not affect bilayer stability, and the tweezer (white dots in *Figure 6F*) was homogenously distributed on the scan area. By contrast, the DOPC/SM/Chol raft mixture appeared as coexisting liquid-disordered ($l_d$) and liquid-ordered ($l_o$) domains, and CLR01 addition induced changes in phase coexistence and decreased height differences between both phases. After 60 min, distinct $l_d$ and $l_o$ domains were no longer visible, and CLR01 was homogenously distributed in the remaining fluid phase. Collectively, these data suggest that CLR01 selectively disrupts heterogeneous lipid-raft enriched membranes.

## CLR01 does not form colloidal aggregates

Inhibitors of amyloid formation may block fibril polymerization by forming colloidal aggregates that sequester peptide via non-specific interactions (*Feng et al., 2008*). Likewise, small molecule aggregates might also interfere non-specifically with viral infection. However, neither inhibition of amyloidogenesis, amyloid remodeling nor inhibition of viral infection by CLR01 were affected by BSA, which would quench small-molecule aggregates (*McGovern et al., 2002*), or by clearance of any potential CLR01 aggregates by centrifugation (*McGovern et al., 2003*) (*Figure 2—figure supplement 4*; *Figure 3—figure supplement 2*; *Figure 5—figure supplements 2, 3, 4*).

To investigate whether CLR01 even forms colloidal aggregates, we performed various tests for CLR01 aggregation using the same buffers and CLR01 concentrations as employed in our biological experiments described above (*Figures 2A–C, 3A–C, 5A*). NMR dilution titrations provided experimental evidence for a very weak CLR01 dimer formation ($K_a$ = ~60 M$^{-1}$) in PBS buffer (10 mM sodium phosphate, 137 mM NaCl, 2.7 mM KCl, pH 7.4), with ~5% CLR01 dimers present when the CLR01 concentration was 500 μM (*Figure 7A*, *Figure 7—figure supplement 1*). This observation is consistent with previously published data (*Dutt et al., 2013*). Dimerization was totally absent in HEPES buffer (25 mM HEPES, 150 mM KOAc, 10 mM Mg(OAc)$_2$, pH 7.4; *Figure 7B*, *Figure 7—figure supplement 2*). Thus, CLR01 is predominantly monomeric.

Diffusion NMR (DOSY) experiments in both buffers revealed CLR01 hydrodynamic radii of ~0.9–1.0 nm, slightly above the monomeric species (*Figure 7C,D*, *Figure 7—figure supplement 3*). Microcalorimetric dilution titrations revealed only minute endothermic heat changes, which argue strongly against an extensive aggregation process (data not shown). Pyrene fluorescence revealed that no critical micelle concentration (cmc) could be determined in a wide concentration range (0–0.5 mM CLR01) encompassing fibril assembly and remodeling conditions (*Figure 7E*). Dynamic light scattering (DLS) experiments showed CLR01 particles with hydrodynamic radius $R_H$ = 5–8 nm in

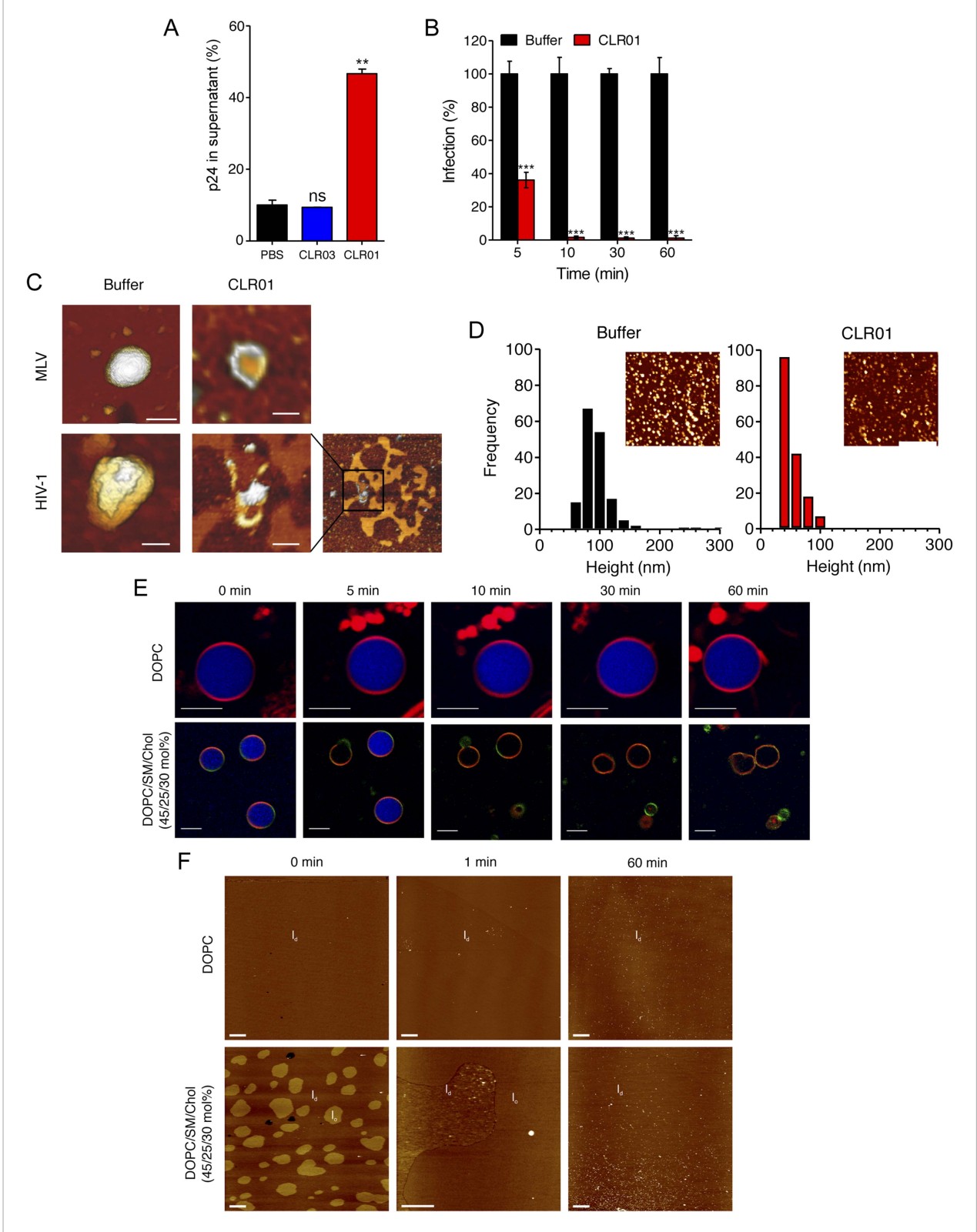

**Figure 6**. CLR01 destroys retroviral particles and selectively disrupts raft-rich membranes. (**A**) CLR01 releases p24 capsid antigen from HIV particles. HIV-1 was incubated with PBS, 100 μM CLR03, or 100 μM CLR01 and centrifuged at 20,000×$g$ and 4˚C for 1 hr. The p24 content of the supernatant was determined via p24 ELISA. Values represent means ±SD. Unpaired t-tests were used to compare the buffer control to the CLR03 or CLR01 condition (ns denotes not significant; ** denotes p < 0.01). (**B**) HIV-1 was incubated at 37˚C with 10 μM CLR01 or buffer control. Aliquots were taken after different
*Figure 6. continued on next page*

*Figure 6. Continued*

time points and analyzed regarding their infectivity using TZM-bl reporter cells. Values represent normalized mean infection rates derived from triplicate measurements ±SD compared to the buffer control (100%). Unpaired t-tests were used to compare the buffer control to the CLR01 condition at each time point (*** denotes p < 0.001). (**C**) CLR01 destroys retroviral particles. Images obtained by atomic force microscopy (AFM) show single MLV and glycoprotein-deficient HIV particles before and after treatment with 100 μM CLR01. Scale bar: 100 nm. (**D**) CLR01 destroys MLV particles. Height distribution of MLV particles after treatment with buffer (left panel) or 100 μM CLR01 (right panel). Values were derived from AFM images shown in the insets. Scale bar: 2 μm. (**E**) CLR01 selectively destroys membranes with high lipid raft content. Giant unilamellar vesicles (GUVs) consisting of pure DOPC were labeled with *N*-Rh-DHPE (red). GUVs containing a mixture of DOPC, SM and Chol (45/25/30 mol%) were labeled with *N*-Rh-DHPE (red) and Bodipy-Chol (green). Both types of GUVs were filled with buffer containing the fluorophore ATTO 647 (blue) and treated with 150 μM CLR01 for the indicated times before images were taken by confocal microscopy. Note that ATTO 647 remains inside the DOPC GUVs treated with CLR01, but escapes the DOPC/SM/Chol GUVs treated with CLR01. Scale bar: 10 μm. (**F**) Upper panel: AFM images (10 μm scans) of a pure DOPC lipid membrane on mica before injection (0 min) and 1 min and 60 min after injection of 800 μl of 150 μM CLR01 in 10 mM $NaH_2PO_4$, pH 7.6 into the AFM fluid cell. The whole scan area is shown with a vertical color scale from dark brown to white corresponding to an overall height of 8 nm. The thickness of the hydrated membrane is 3.7 nm. Lower panel: AFM image (10 μm scan) of a DOPC/SM/Chol (45/25/30 mol%) lipid membrane on mica before injection (0 min) and 1 min and 60 min after injection of 800 μl of 150 μM CLR01 in 10 mM $NaH_2PO_4$, pH 7.6 into the AFM fluid cell. The whole scan area is shown with a vertical color scale from dark brown to white corresponding to an overall height of 8 nm and indicating a homogeneous lipid bilayer with coexisting domains in $l_o$ (liquid-ordered) and $l_d$ (liquid-disordered domain) phase. The height difference between domains is 1 nm; the $l_d$ phase has a thickness of 4.0 nm.

PBS at concentration between 200 and 1000 μM, plus a very minor fraction of particles with a $R_H$ = 20–40 nm (*Figure 7F*; data not shown). We did not observe CLR01 particles with a $R_H$ of ~95–400 nm, which is the size range typically associated with colloidal small-molecule aggregates (*McGovern et al., 2002*). No CLR01 particles could be detected at 10 or 50 μM in PBS (data not shown). In HEPES buffer, no particles were detected at any of the concentration tested (between 10 and 1000 μM) (*Figure 7F*; data not shown). The ~0.9–1 nm hydrodynamic radius of CLR01 revealed by DOSY (*Figure 7C,D*) is below the detection limit of our DLS instrument, and hence, we cannot resolve the CLR01 monomer (*Figure 7F*). Importantly, DLS overemphasizes large particles because the scattered-light intensity is proportional to the square of the particle mass. The scattering intensity in the solutions of CLR01 (1 mM) in PBS was ~3% that of similar samples containing $Mg_3(PO_4)$ colloids (10 mM $Mg_3(PO_4)$) or SDS micelles (2% SDS (wt/vol)), suggesting that the observed species represented a small fraction of the CLR01 molecules (data not shown). Thus, under all assay concentrations the vast majority of the tweezer molecules are monomeric. Minute amounts of dimers or higher order species may be present at high concentrations of CLR01 in some cases in PBS (fibril assembly buffer) but are absent from the HEPES buffer (fibril remodeling buffer). Thus, it is highly unlikely that colloidal CLR01 aggregates contribute to the observed activity of the tweezer.

## CLR01 exhibits broad antiviral activity against enveloped viruses

CLR01 exhibited a surprising ability to disrupt viral membranes (*Figures 5A, 6A,C–F*), but did not affect cell viability like various non-ionic and anionic surfactants (*Figure 4—figure supplement 3*). We therefore explored whether CLR01 could act as a general inhibitor of enveloped viruses. To test this concept, human cytomegalovirus (HCMV), herpes simplex virus type 2 (HSV-2), and hepatitis C virus (HCV) were treated with CLR01 or CLR03 and then assessed for their ability to infect target cells. Remarkably, CLR01, but not CLR03, reduced infection rates of all three analyzed enveloped viruses (*Figure 8A–C*). By contrast, CLR01 did not inhibit infection by the non-enveloped human adenovirus type 5 (HAdV5) (*Figure 8D*). Thus, the tweezer is a broad-spectrum inhibitor of enveloped viruses, including viruses that can be sexually transmitted such as HIV-1, HSV-2, and HCV.

## CLR01 antagonizes the infection-enhancing property of human semen

CLR01 binds exposed lysine and arginine residues in amyloidogenic peptides (*Sinha et al., 2011*; *Attar et al., 2012*; *Prabhudesai et al., 2012*; *Sinha et al., 2012*; *Acharya et al., 2014*; *Ferreira et al., 2014*; *Zheng et al., 2015*). In vivo, the amyloids we examined are present in the complex environment of human semen. We wondered whether the interactions between CLR01 and seminal peptides might be hindered in conditions resembling those in seminal fluid. To test whether this is the case, lyophilized PAP248-286 was dissolved in an artificial semen simulant (AS) (*Owen and Katz, 2005*; *Olsen et al., 2012*) containing 50 mg/ml BSA and agitated until fibril formation was complete.

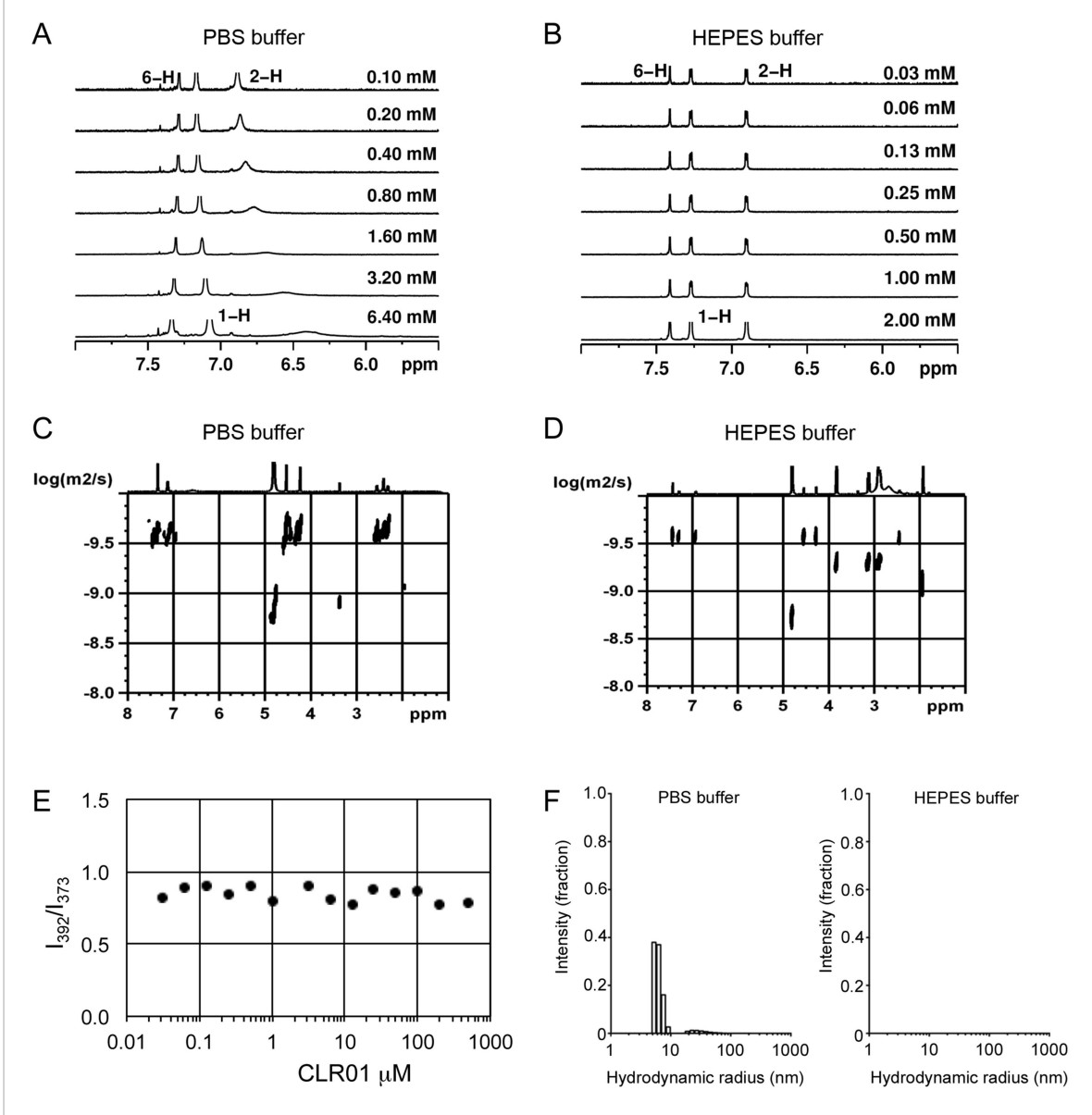

**Figure 7.** CLR01 does not form colloid or micelle aggregates. (**A**) [1]H NMR dilution titration of CLR01 from 6 mM to 100 μM in PBS leads to very weak dimerization; (**B**) in HEPES buffer CLR01 remains monomeric over the entire concentration range. (**C**) DOSY spectrum of 6 mM CLR01 in PBS buffer; cross peaks at 4.8 ppm correspond to aqueous solvent. (**D**) DOSY spectrum of 2 mM CLR01 in HEPES buffer; cross peaks between 2 and 4 ppm correspond to HEPES buffer, cross peaks at 4.8 ppm correspond to the aqueous solvent. (**E**) Attempted cmc determination by analysis of the $I_{392}/I_{373}$ emission intensity ratio of 10 μM PBS-buffered pyrene solutions containing CLR01 from 30 nM to 500 μM produces a straight line, as opposed to the SDS control (data not shown). (**F**) DLS measurements of CLR01 (1 mM) in PBS or HEPES buffer show formation of minute amounts of particles with $R_H$ ~5–8 nm or ~20–40 nm in PBS. No particles were detected in HEPES buffer. The vast majority of CLR01 molecules are too small to be detected and are presumably monomeric.

The following figure supplements are available for figure 7:

**Figure supplement 1**. Aromatic proton shifts during dilution of PBS-buffered CLR01.

**Figure supplement 2**. Aromatic proton shifts during dilution of HEPES-buffered CLR01.

**Figure supplement 3**. Stejskal and Tanner Plot.

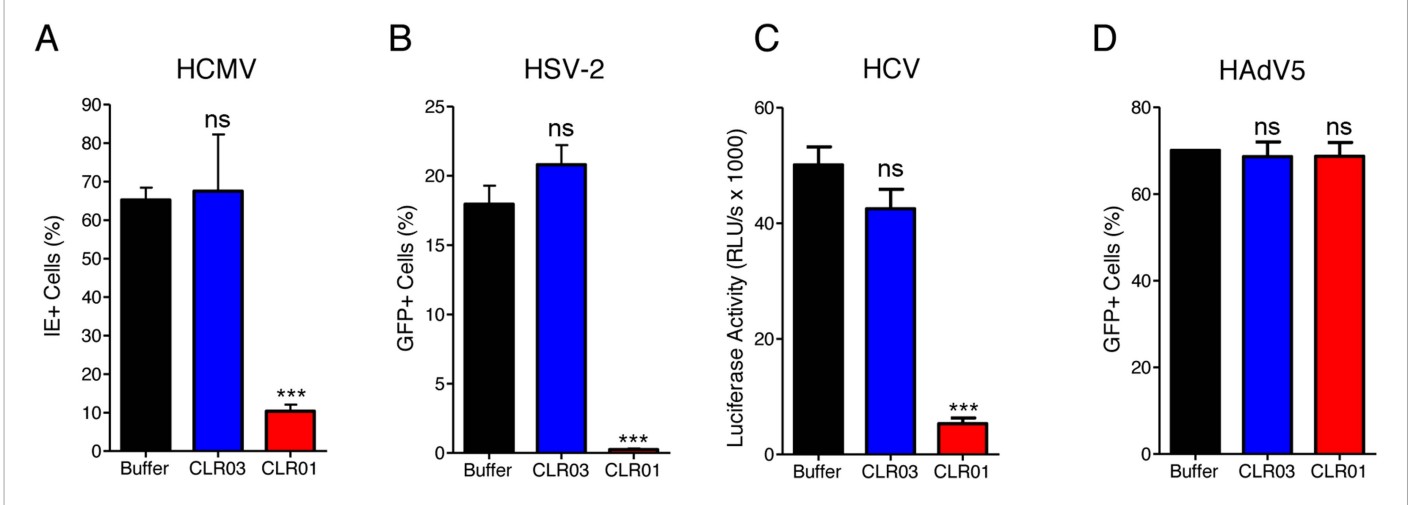

**Figure 8**. CLR01 is a broad-spectrum inhibitor of enveloped viruses. (**A**) Human cytomegalovirus was incubated with PBS, 100 µM CLR03, or 100 µM CLR01. Afterwards, HFF cells were infected and immediate early (IE) antigen positive cells were counted 1 day post infection as a measure for infectivity. Values are means ±SD (n = 3). Unpaired t-tests were used to compare the buffer control to the CLR03 or CLR01 condition (ns denotes not significant; *** denotes p < 0.001). (**B**) Herpes simplex virus type 2 comprising a GFP reporter gene was treated with PBS, 100 µM CLR03 or 100 µM CLR01 and added to Vero cells. GFP-positive cells were counted using flow cytometry 2 days post infection. Values represent means ±SD (n = 3). Unpaired t-tests were used to compare the buffer control to the CLR03 or CLR01 condition (ns denotes not significant; *** denotes p < 0.001). (**C**) A luciferase encoding hepatitis C virus was treated with 150 µM CLR01 or 150 µM CLR03 and used for infection of Huh-7.5 reporter cells. Infection was measured 3 days post infection. Values represent means ±SEM (n = 3). Unpaired t-tests were used to compare the buffer control to the CLR03 or CLR01 condition (ns denotes not significant; *** denotes p < 0.001). (**D**) A GFP-reporter adenovirus type 5 was added to A549 cells after treatment with 158 µM CLR01 or 158 µM CLR03. GFP positive cells were counted using flow cytometry 1 day post infection. Values represent means ±SD (n = 3). Unpaired t-tests were used to compare the buffer control to the CLR03 or CLR01 condition (ns denotes not significant).

These PAP248-286(AS) fibrils were then diluted in AS and incubated with CLR01. A reduction in ThT fluorescence intensity to 43% of the initial value was detected (*Figure 9A*). This finding confirms that CLR01 maintains its amyloid-remodeling activity in a complex solution resembling seminal fluid.

Finally, we tested whether the molecular tweezer inhibited semen-mediated infection enhancement, as described (*Müller and Münch, 2016*). Seminal plasma, which was obtained from pooled semen of 10 donors, was treated with CLR01 or CLR03. Thereafter, a CCR5-tropic lab-adapted virus or a transmitted/founder virus were incubated with 10% of these solutions followed by infection of TZM-bl cells. Consistent with previous studies (*Münch et al., 2007*; *Hauber et al., 2009*; *Roan et al., 2009*; *Kim et al., 2010*), seminal plasma enhanced infection of both viruses by 10- or 8-fold, respectively (*Figure 9B*). CLR01 decreased viral infectivity enhancement in a concentration-dependent manner, while CLR03 had no effect on semenmediated infection enhancement (*Figure 9B*). At a CLR01 concentration of 31 µM, any stimulatory effect of semen was abolished (*Figure 9B*). Thus, CLR01 prevents HIV infection in the presence of semen, the main vector for viral transmission in the human population.

## Discussion

Despite the development of several different classes of topical microbicides, none have proven safe and effective at HIV prevention. The failure of topical microbicide candidates in previous clinical trials has been attributed to lack of adherence (*Van Damme et al., 2008*; *Marrazzo et al., 2015*), adverse effects (*McGowan et al., 2011*), and a greatly diminished antiviral efficacy in the presence of semen (*Neurath et al., 2006*; *Zirafi et al., 2014*). This bodily fluid is not only the main vector for HIV transmission but also contains cationic amyloid fibrils that markedly increase viral infectivity (*Castellano and Shorter, 2012*; *Münch et al., 2014*). Various biological polyanions such as heparin or other glycosaminoglycans can prevent these cationic fibrils from enhancing HIV infection (*Roan et al., 2009*). Unfortunately, however, such anionic polymers have been unsuccessful in past clinical microbicide trials due to their poor bioavailability and induction of inflammatory responses in the

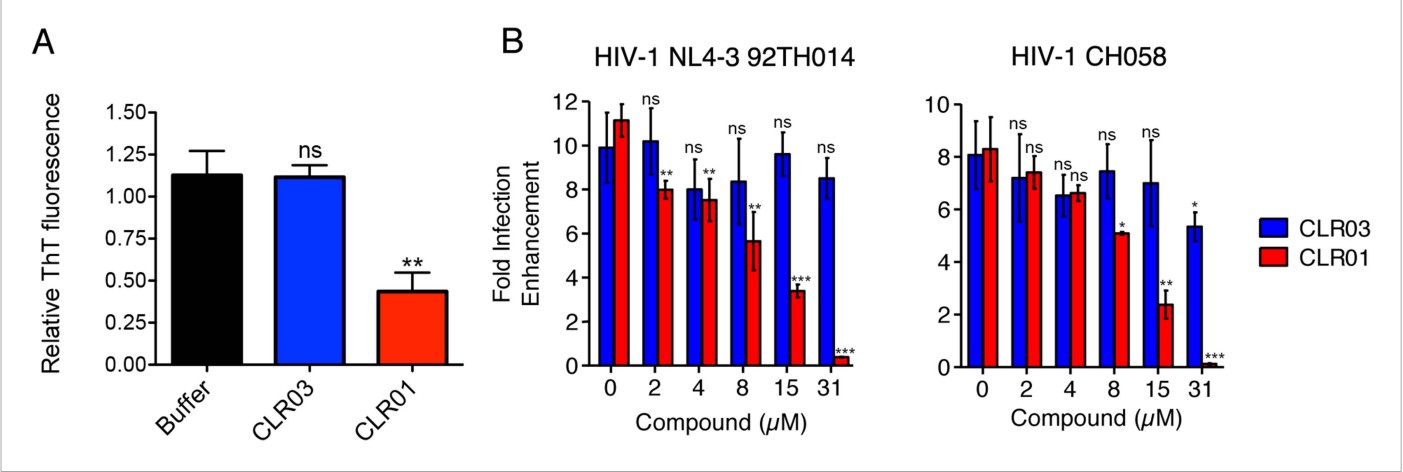

**Figure 9**. CLR01 diminishes the infection enhancing property of semen. (**A**) SEVI fibrils were formed in an artificial semen simulant (AS) (*Owen and Katz, 2005*). The resulting fibrils were then diluted to 20 µM in AS and treated with 200 µM CLR01 or CLR03, or with buffer for 2 hr. Fibril integrity was assessed using ThT fluorescence. Values represent means ±SEM (n = 4). A one-way ANOVA with the post hoc Dunnett's multiple comparisons test was used to compare the buffer alone control to the other conditions (ns denotes not significant; ** denotes p < 0.01). (**B**) CLR01 abrogates semen-mediated enhancement of HIV infection. Seminal plasma (10%) or cell culture medium containing CLR01 or CLR03 was mixed with CCR5-tropic HIV-1 or transmitter/founder HIV-1 CH058. After 10 min, TZM-bl cells were infected and infectivity was measured 3 day post infection. Shown are the n-fold increased infection rates obtained for semen-treated virus relative to those of medium-treated virus. Values represent means ±SD (n = 4). Unpaired t-tests were used to compare the buffer control (0 µM compound) to the CLR03 or CLR01 condition at each concentration (ns denotes not significant; * denotes p < 0.05; ** denotes p < 0.01; *** denotes p < 0.001).

genital tract, which actually augment HIV transmission by recruiting HIV-susceptible target cells to the genital mucosa (*Lüscher-Mattli, 2000*; *van de Wijgert and Shattock, 2007*). We have suggested that future microbicide endeavors should focus on agents that simultaneously and safely target HIV and the host factors that are exploited by the virus to facilitate its transmission (*Castellano and Shorter, 2012*; *Zirafi et al., 2014*; *Roan and Münch, 2015*). Here, we report that CLR01, a lysine- and arginine-specific molecular tweezer (*Fokkens et al., 2005*), not only counteracts the infection-enhancing activity of seminal amyloids and semen, but also directly destroys HIV virions (*Figure 10*). CLR01 is a highly promising topical microbicide candidate because it possesses potent antiviral and anti-amyloid activity, displays minimal toxicity in vivo (*Prabhudesai et al., 2012*; *Attar et al., 2014*; *Ferreira et al., 2014*), and is efficacious in human seminal fluid.

It was surprising that CLR01 not only affected the formation and function of seminal amyloids but also displayed a broad and direct antiviral activity against HIV and other enveloped viruses (*Figure 8*). These diverse activities made us concerned that CLR01 might disrupt amyloidogenesis and viral infection via a non-specific mechanism involving the formation of colloidal CLR01 aggregates (*McGovern et al., 2002*, *2003*; *Shoichet, 2006*; *Feng et al., 2008*). Nevertheless, multiple lines of evidence argue against a non-specific, colloidal mechanism of CLR01 activity. First, using a variety of biophysical techniques, we were unable to detect significant quantities of colloidal CLR01 aggregates under the conditions employed in our experiments (*Figure 7*). Second, the anti-amyloid and antiviral activity of CLR01 was unaffected by high concentrations of BSA (*Figure 2—figure supplement 4*; *Figure 3—figure supplement 2*; *Figure 5—figure supplement 2 and 3*; *Figure 9A*), which would adsorb and quench the activity of any potential colloidal micelles (*McGovern et al., 2002*; *Feng et al., 2008*). Third, attempts to preclear CLR01 solutions by centrifugation to remove any colloidal aggregates (*McGovern et al., 2003*; *Feng et al., 2008*) had no effect on CLR01 anti-amyloid or antiviral activity (*Figure 2—figure supplement 4*; *Figure 3—figure supplement 2*; *Figure 5—figure supplement 4*). Fourth, the shallow CLR01 dose–response curves are not typical of colloidal aggregate inhibitors, which typically exhibit steep dose–response curves (*Shoichet, 2006*). Finally, a feature of small molecules that form colloidal aggregates is that they are non-specific inhibitors (*McGovern et al., 2002*, *2003*; *Feng et al., 2008*). By contrast, CLR01 displays specificity for amyloidogenesis by peptides that harbor lysine and arginine residues. Thus, if the lysine and

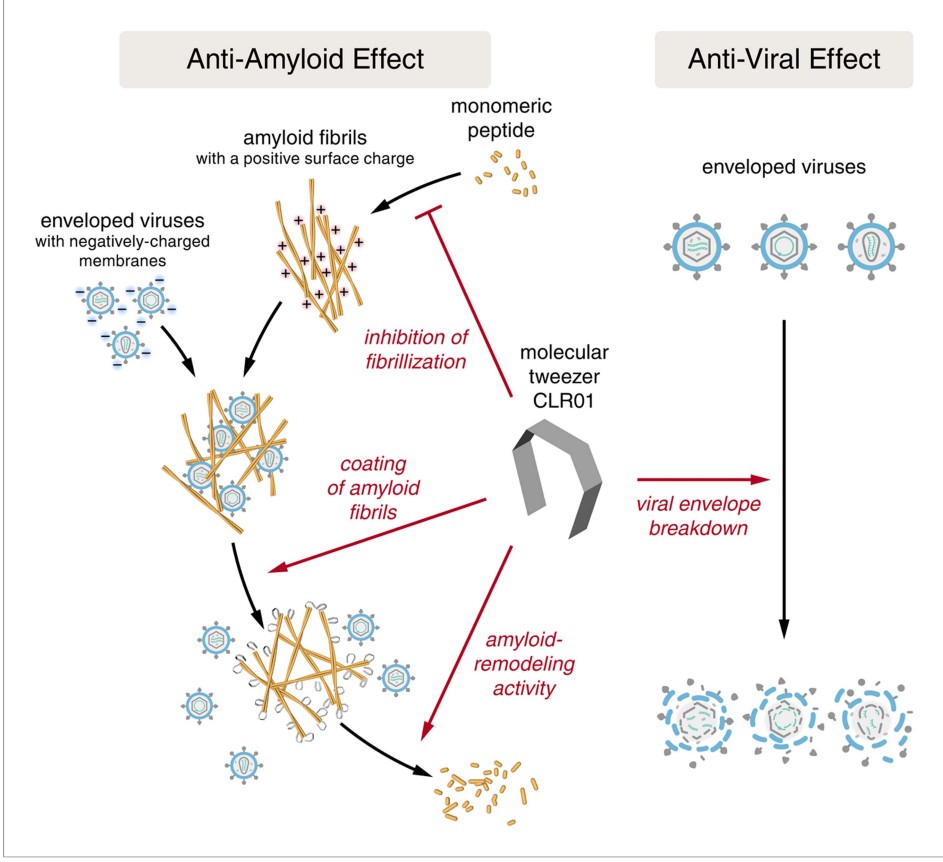

**Figure 10**. CLR01 acts as a dual-function inhibitor of viral infection. Schematic overview of the anti-amyloid and antiviral effects of CLR01.

arginine residues of PAP248-286 or PAP85-120 are replaced with alanine, then CLR01 can neither inhibit fibril assembly nor remodel preformed fibrils (*Figures 2G–I, 3I–L*). Likewise, the antiviral activity of CLR01 is restricted to enveloped viruses and CLR01 is ineffective against non-enveloped viruses (*Figure 8*). This specificity is inconsistent with an indiscriminate, non-specific mechanism of action mediated by colloidal CLR01 aggregates. Importantly, the anti-amyloid and antiviral activity of CLR01 could be eliminated by excess lysine or poly-L-lysine. Thus, the lysine-binding cavity of CLR01 plays an important role in both anti-amyloid and antiviral activity.

The ability of CLR01 to antagonize seminal amyloid in the presence of BSA and to prevent HIV infection in the presence of seminal fluid (*Figure 9*) raises another question. Why is CLR01 activity not quenched by lysine or arginine residues on the surface of BSA or on other proteins abundant in seminal fluid? Here, it is important to consider the profound difference between freely accessible lysines on intrinsically unstructured domains, which can all be complexed by the tweezers, and lysines in folded proteins whose surface often prohibits the sterically demanding tweezer molecule to approach a lysine residue close enough for the threading mechanism (*Figure 1B*). The interaction between free lysine and CLR01 has a $K_d$ of ~10 µM, with fast on and off rates (*Fokkens et al., 2005*; *Bier et al., 2013*; *Dutt et al., 2013*). On folded proteins, however, only a few lysines are accessible for the relatively large molecular skeleton of the tweezers. For example, only 5 of 17 surface lysines in a 14-3-3 protein are most likely to be complexed by CLR01 (*Bier et al., 2013*). Even in an unstructured small protein, such as Aβ, CLR01 has clear preference for the free Lys16 over Lys28 (*Sinha et al., 2011*) because the latter is engaged in stabilizing hydrophobic interactions and salt bridges (*Petkova et al., 2002*; *Lazo et al., 2005*; *Xiao et al., 2015*). Moreover, because of the high on and off rates, CLR01 is expected to only disrupt relatively weak interactions as well as events that depend on critical and accessible lysine or arginine residues. Therefore, the tweezer has shown selectivity in multiple cases

that on first impression may appear counter-intuitive. For example, inhibition of abnormal protein aggregation is typically achieved at ∼1:1–1:5 protein:CLR01 concentration (*Sinha et al., 2011, 2012*; *Acharya et al., 2014*), whereas disruption of controlled protein polymerization, for example, of tubulin, requires ∼55-fold excess CLR01 (*Attar et al., 2014*) and enzyme inhibition requires ∼1000-fold excess CLR01 (*Talbiersky et al., 2008*). Moreover, in vivo, CLR01 shows no toxicity at doses 2–3 orders of magnitude above those needed to enable clearance of amyloid (*Attar et al., 2012*; *Prabhudesai et al., 2012*; *Attar et al., 2014*; *Ferreira et al., 2014*). Thus, selectivity for lysine-bearing amyloid conformers in complex biological fluids is achieved by CLR01.

We suggest that the lysine-rich nature of seminal amyloid peptides (*Figure 1C–E*) makes them unusually sensitive to CLR01 activity. Indeed, substoichiometric CLR01 concentrations (relative to peptide monomers) suffice to inhibit PAP248-286, PAP85-120, and SEM1(45-107) fibrillization (*Figure 2D–F*). Remarkably, substoichiometric CLR01 concentrations (relative to peptide monomers) also sufficed to effectively remodel SEVI and PAP85-120 fibrils (*Figure 3G,H*). Curiously, SEM1(45-107) fibrils could not be remodeled by CLR01. Why SEM1(45-107) fibrils were refractory to CLR01 remains unclear and requires further study. CLR01 may require longer time periods to remodel SEM1(45-107) fibrils as with remodeling of Aβ fibrils, which required several weeks (*Sinha et al., 2011*). It will also be important to determine whether some lysine and arginine residues in PAP248-286, PAP85-120, and SEM1(45-107) are more critical than others for CLR01 activity.

How does CLR01 directly antagonize viral infection? Since the CLR03 control, which has a similar negative charge as CLR01 (*Sinha et al., 2011*), was entirely ineffective at blocking viral infection, a purely polyanion-like mechanism that prevents the interaction of virions and cells by increasing charge repulsions (*Neurath et al., 2002*; *Weber et al., 2005*) could be excluded. Instead, we demonstrate that CLR01 selectively disrupts membranes containing elevated levels of sphingomyelin and cholesterol (*Figure 6*). Envelopes of HIV-1, herpes viruses, and HCV differ significantly from the cellular plasma membrane as they are more highly enriched in these and other lipids (*van Genderen et al., 1994*; *Brügger et al., 2006*; *Chan et al., 2008*; *Lorizate et al., 2009, 2013*; *Merz et al., 2011*). Indeed, the HIV membrane possesses a highly unusual lipid composition, which is distinct even from the detergent-resistant microdomains typically found in the plasma membrane, indicating that HIV budding involves a specific lipid raft clustering process (*Brügger et al., 2006*). For example, although HIV membrane resembles lipid raft microdomains in that it is enriched for saturated lipids, phosphatidylserine, plasmalogen-phosphatidylethanolamine, cholesterol, and sphingolipids, it is also enriched for the unusual sphingolipid dihydrosphingomyelin (*Brügger et al., 2006*; *Lorizate et al., 2009, 2013*). The unique lipid composition of viral membranes may render them uniquely sensitive to CLR01.

Importantly, unlike various non-ionic and anionic surfactants, CLR01 has minimal effects on cell viability (*Figure 4—figure supplement 3*). Thus, CLR01 might leverage a therapeutic opportunity provided by the physiological difference between static viral membranes and biogenic cellular membranes capable of self-repair. Static viral membranes are unable to withstand disruption by CLR01, whereas cells can likely repair lipid-raft-rich regions of the plasma membrane that might be disrupted by CLR01. In this regard, CLR01 may possess similarities to other broad-spectrum antivirals such as LJ001, a lipophilic aryl methyldiene rhodanine derivative, which selectively disrupt diverse viral membranes but are ineffective against non-enveloped viruses, and also do not disrupt cell membranes (*Wojcechowskyj and Doms, 2010*; *Wolf et al., 2010*; *Vigant et al., 2013*). We investigated whether the tweezer might recognize the charged lipid head groups but found via NMR chemical-shift experiments that this was not the case (data not shown). However, experiments on various phase boundaries strongly indicated that the tweezer could migrate into lipid bilayers due to its amphiphilic character (data not shown). A pronounced destabilizing effect may be triggered if this happens at the edge of lipid rafts that are enriched in virion membranes. Importantly, HSV-2, HCMV, and HCV can also be transmitted via sexual intercourse (*Handsfield et al., 1985*; *Rapp, 1989*; *Tedder et al., 1991*). Thus, a CLR01-based microbicide may not only protect from HIV-1 acquisition but also from other major human pathogens.

CLR01 binding to lysine residues could potentially disrupt normal protein function leading to toxicity or side effects. However, CLR01 showed no signs of cytotoxicity in this study and has previously been safely employed in multiple cell and animal models (*Sinha et al., 2011*; *Attar et al., 2012*; *Prabhudesai et al., 2012*; *Attar et al., 2014*). For example, transgenic mice that were given 10 mg CLR01 a day per 1 kg body weight intraperitoneally for 30 days showed no apparent toxicity or adverse effects (*Attar et al., 2014*). Moreover, it is conceivable that the application of CLR01 on mucosal surfaces should be even safer compared to its systemic administration.

Unlike previous microbicide candidates that block HIV infection by a single mechanism, a CLR01-based formulation would interfere with various essential steps in sexual HIV transmission (*Figure 10*). On a minute time frame, the tweezer destroys infectious HIV particles that are present in semen, prevents the formation of amyloid–virus complexes by neutralization of the fibril surface charge, and also displaces pre-bound virions from fibrils (*Figure 10*). On a longer time frame (hours), CLR01 not only prevents the formation of infection-enhancing seminal amyloids upon ejaculation but also remodels fibrils that are already abundant in semen after ejaculation or formed during semen liquefaction (*Figure 10*). Given the time frame of these effects, we suggest that the primary mechanism of CLR01 action is via its direct effect on the virus, and that the anti-amyloid effects, although potent, play a secondary role. Regardless, these combined antiviral and anti-amyloid activities and the encouraging safety data make CLR01 a promising broad-spectrum, topical microbicide against HIV-1 and other sexually transmitted viruses. We anticipate that large-scale synthesis of CLR01 could be relatively inexpensive (less than $1 per mg), which would enable facile development as a broad-spectrum antiviral microbicide.

## Materials and methods

### Computational details

The interactions of PAP248-286 with the molecular tweezer CLR01 and the spacer CLR03 were investigated using Replica Exchange Molecular Dynamics (REMD) simulations (*Sugita and Okamoto, 1999*; *Okabe et al., 2001*) performed with Gromacs 4.6 (*Hess et al., 2008*) and the CHARMM27 force field (*Mackerell et al., 2004*; *Bjelkmar et al., 2010*). The water solvent was treated explicitly using the TIP3P model (*Jorgensen et al., 1983*). The parameters for CLR01 and CLR03 were obtained using the Swissparam server and tested previously (*Zoete et al., 2011*; *Bier et al., 2013*). The initial coordinates of PAP248-286 were taken from the Protein Data Bank, code 2L3H (*Nanga et al., 2009*). The temperature range for the REMD simulations was 290–330 K. The temperature distribution was obtained as described by *Patriksson and van der Spoel (2008)*. Here, we focused our discussion on the 300 K trajectory. Three systems were investigated: (1) PAP248-286, (2) PAP248-286 with 7 CLR01 molecules (one for each Lys or Arg residue, with the exception of Lys272 which was not sterically accessible in the initial structure but became available during the simulations), and (3) PAP248-286 with 8 CLR03 molecules interacting via the hydrogen phosphate groups with each Lys or Arg residue of PAP248-286. For each system, 22 replicas were simulated during 60 ns. The cluster analysis of the REMD simulations was performed with the gromos method using a cut-off value of 0.2 nm (*Daura et al., 1999*).

### Small molecules, peptides, and seminal plasma

All chemicals or biochemicals were from Sigma–Aldrich (St. Louis, MO) unless otherwise stated. CLR01 and CLR03 were generated as described previously (*Fokkens et al., 2005*), and 7.4–20 mM stock solutions were prepared in water or in PBS. Synthetic peptides PAP248-286, PAP248-286(Ala), PAP85-120, and SEM1(45-107) or (49-107) were purchased from Keck Biotechnology Resource Laboratory (Yale University, New Haven, CT) or Celtek peptides (Franklin, TN). PAP85-120(Ala) was purchased from Bachem (King of Prussia, PA). For fibril formation, peptides were reconstituted and assembled into fibrils as previously described (*Münch et al., 2007*; *Roan et al., 2011*; *Arnold et al., 2012*). L-lysine, poly-L-lysine hydrobromide (molecular weight 4000–15,000 by viscosity), and poly-L-lysine hydrobromide (molecular weight 70,000–150,000 by viscosity) were from Sigma–Aldrich. Seminal plasma represents the cell-free supernatant fraction of pooled human semen centrifuged at 20,000×g for 30 min at 4°C.

### [1]H NMR dilution titration

CLR01 was dissolved in PBS (10 mM sodium phosphate, 137 mM NaCl, 2.7 mM KCl, pH 7.4) or HEPES buffer (25 mM HEPES, 150 mM KOAc, 10 mM Mg(OAc)$_2$, pH 7.4) and diluted from 6.4 mM to 100 µM. In the corresponding [1]H NMR spectra, the chemical shift changes of the inner aromatic proton forming a binding isotherm, were analyzed by nonlinear regression and furnished the 1:1 dimerization constant.

### DOSY experiments

CLR01 was dissolved in PBS or HEPES buffer (pH 7.4) and diluted from 6.4 mM to 200 µM. DOSY spectra were measured for all concentrations of CLR01, and the diffusion coefficient was determined

from the corresponding Stejskal and Tanner Plot. The diffusion coefficient was converted into the hydrodynamic radius by way of the Stokes–Einstein equation.

## Isothermal titration calorimetry (ITC) experiments

A dilution experiment was performed with a HEPES buffered solution of CLR01 (6 mM–100 µM). Evolved heats were very small (0.1–0.2 µcal per injection) and endothermic.

## Critical micelle concentration (cmc) determination

Pyrene (10 µM) was dissolved in PBS or HEPES buffer containing 5% of DMSO, and the intensity ratio of both fluorescence emission maxima $I_{392}/I_{373}$ was determined at various CLR01 concentrations (10–500 µM). As a reference, SDS solutions in the same buffer were analyzed at 0.1–50 mM (cmc at 6–8 mM).

## Dynamic light scattering (DLS) experiments

CLR01 was dissolved in phosphate-buffered saline (PBS 10 mM sodium phosphate, 137 mM NaCl, 2.7 mM KCl, pH 7.4) or 25 mM HEPES, 150 mM KOAc, 10 mM Mg(OAc)$_2$, pH 7.4 at 10, 50, 200, 500, or 1000 µM. The solutions were transferred to DLS cuvettes and centrifuged for 30 min at 5000×$g$ to pellet dust particles. Solutions were measured using an in-house-built system with a He-Ne laser, model 127 (wavelength 633 nm, power 60 mW; Spectra Physics Lasers, Mountain View, CA). Light scattered at 90° was collected using image-transfer optics and detected by an avalanche photodiode built into a 256-channel PD2000DLS correlator (Precision Detectors, Bellingham, MA). The size distribution of scattering particles was reconstructed from the scattered light correlation function using PrecisionDeconvolve software (Precision Detectors) based on the regularization method by Tikhonov and Arsenin (*Tikhonov and Arsenin, 1977*). As a reference, Mg$_3$(PO$_4$) colloids (10 mM Mg$_3$(PO$_4$)) or SDS micelles (2% SDS (wt/vol)) were analyzed.

## Fibril assembly and remodeling assays

For assembly experiments, each reconstituted peptide was incubated with CLR01 or CLR03 and agitated at 37°C at 1400 rpm. At various time points, aliquots (1 µl) were removed and added to 25 µM ThT in PBS (200 µl). Changes in fluorescence (excitation: 440 nm, emission: 482 nm) were measured using a Tecan Safire2 microplate reader (Tecan, Männedorf, Switzerland). To assess the extent of fibril assembly using sedimentation analysis, fibril samples (20 µM fibrils, 100 µl volume) were centrifuged for 10 min at 13,200 rpm. The supernatant was carefully removed and transferred to a new tube, and the pellet was redissolved in an equal volume (100 µl) of buffer. Then, 50 µl of 3× sample buffer (6% SDS, 187.5 mM Tris, 30% glycerol, 10% β-mercaptoethanol, 0.05% bromophenol blue, pH 6.8) was added to the supernatant and pellet fractions. Samples were analyzed by SDS-PAGE using 10–20% Tris-Tricine peptide gels and XT Tricine running buffer (Bio-Rad, Hercules, CA) and visualized by coomassie staining. A gradient of soluble peptide controls were also run on the gels. Densitometry (using Image J software) was used to quantify the percent of protein in the pellet fractions by comparing to a standard curve created from the soluble peptide controls. For amyloid-remodeling experiments, fibrils (20 µM, based on peptide monomer concentrations) were diluted into an assay buffer (25 mM HEPES, 150 mM KOAc, 10 mM Mg(OAc)$_2$, pH 7.4) in the presence of ATP (5 mM) and incubated with either CLR01 or CLR03. Aliquots (5 µl) were removed at various time points and added to 25 µM ThT in PBS (55 µl). ThT fluorescence was measured as above. The artificial semen simulant was prepared as described previously (*Owen and Katz, 2005*; *Olsen et al., 2012*), and is comprised of: 18 mM citrate; 40 mM chloride; 7 mM calcium; 4.5 mM magnesium; 28 mM potassium; 220 mM sodium; 2 mM zinc; 15 mM fructose; 6 mM glucose; 50.4 mg/ml BSA; 7 mM lactic acid; 7.5 mM urea; all in a 123 mM sodium phosphate base, pH 7.7. In some fibril assembly and remodeling experiments, BSA (10 mg/ml) was also included. To assess any contribution of potential colloidal CLR01 aggregates, CLR01 was first reconstituted at the requisite concentration for fibril assembly or remodeling reactions, and then centrifuged at 16,100×$g$ for 20 min at 25°C. The supernatant fraction was then collected and used in fibril assembly or remodeling reactions. For some fibril assembly and remodeling reactions, CLR01 was incubated with a 200-fold molar excess of L-lysine or 10-fold molar excess of poly-L-lysine (molecular weight 4000–15,000 by viscosity) for 10 min on ice prior to addition to unassembled peptide or preformed fibrils. To determine IC$_{50}$ or EC$_{50}$ values and Hill slopes for dose–response relationships, the

data were analyzed using GraphPad Prism software. A nonlinear regression analysis (log(inhibitor) vs response–variable slope) was used and fitted with the least squares (ordinary) fit.

## ThT displacement assay

Preformed SEVI, PAP85-120, or SEM1(45-107) fibrils (5 µM monomer) were preincubated with ThT (25 µM) for 30 min at room temperature. Buffer (PBS), CLR01 (250 µM), or a known competitor of ThT binding, BTA-1 (250 µM) (Lockhart et al., 2005) were then added and incubated for 10 min at room temperature. ThT displacement was then assessed by fluorescence measurements.

## Transmission electron microscopy

Aliquots were removed from the assembly and remodeling reactions, spotted for 10 min on Formvar carbon-coated grids (EM Sciences), stained for 5 min with 2% uranyl acetate, and washed with distilled water. Samples were visualized using a Jeol-1010 transmission electron microscope.

## Circular dichroism spectroscopy

CD spectra were collected on an AVIV Model 410 Circular Dichroism Spectrometer. Mean residue ellipticity (MRE) was calculated using the equation $MRE = \theta/(10lcN)$ where $\theta$ is the measured ellipticity in millidegrees, l is the pathlength in cm, c is the molar protein concentration, and N is the number of residues.

## Zeta potential

PAP248-286, PAP85-120, and SEM1(49-107) fibrils were treated with a 10-fold excess of CLR01 or CLR03. After centrifugation at 20,000×g for 10 min, the pellets were resuspended in 1 mM KCl. Zeta potential was measured using the Zeta Nanosizer (Malvern Instruments, UK). To analyze the impact of lysine or poly-L-lysine on the binding of CLR01 to semen amyloids, SEVI fibrils were treated with a twofold excess of CLR01 and increasing concentrations of lysine (1, 10 and 100-fold excess over CLR01) or poly-L-lysine (molecular weight 70,000–150,000 by viscosity; 0.02, 0.25 and 2.5-fold excess of lysine monomer equivalents over CLR01). After centrifugation at 20,000×g for 10 min, samples were processed and measured as described above.

## Confocal microscopy

Fibrils (200 µg/ml in PBS) were stained with Proteostat Amyloid Plaque Detection Kit (Enzo Life Sciences, Plymouth Meeting, PA). Fibrils were then treated with 20-fold excess CLR01 or CLR03 and mixed 1:2 with MLV-Gag-YFP virions. Samples were transferred to µ-slides VI0.4 (Ibidi, Munich, Germany) and imaged with a Zeiss LSM confocal microscope.

## Effect of CLR01 on amyloid and semen-mediated enhancement of HIV infection

The reporter cell line TZM-bl was obtained through the NIH ARRRP and cultured in cell culture medium (DMEM medium supplemented with 120 µg/ml penicillin, 120 µg/ml streptomycin, 350 µg/ml glutamine and 10% inactivated fetal calf serum (FCS), Gibco, Life Technologies, Frederick, MD). This cell line is stably transfected with an LTR-lacZ cassette and expresses CD4, CXCR4, and CCR5. Upon infection with HIV-1, the viral protein Tat is expressed which activates the long terminal repeat (LTR) resulting in the generation of β-galactosidase molecules.

Virus stocks of X4-tropic HIV-1 NL4-3, R5-tropic HIV-1 NL4-3 92TH014, and of the transmitter/founder viruses THRO.c and CH058.c (kindly provided by B Hahn) were generated by transient transfection of 293T cells as described (Münch et al., 2007). After transfection and overnight incubation, the transfection mixture was replaced with 2 ml cell culture medium with 2% inactivated FCS. After 40 hr, the culture supernatant was collected and centrifuged for 3 min at 330×g to remove cell debris. Virus stocks were analyzed by p24 antigen ELISA and stored at −80°C.

To assess the effect of CLR01 and CLR03 on amyloid-mediated enhancement of HIV-1 infection, $10^4$ TZM-bl cells in 180 µl cell culture medium were seeded in 96-well flat-bottom plates the day before infection. 200 µg/ml fibrils (44 µM SEVI, 45 µM PAP85-120 fibrils, 30 µM SEM1(49-107) fibrils) were treated with a 20-fold molar excess of CLR01 or CLR03 for 10 min at room temperature, serially diluted fivefold and then mixed with R5-tropic HIV-1 NL4-3 92TH014 (0.5 ng/ml p24 antigen).

After 5 min, 20 µl of these mixtures were added to TZM-bl cells and infection rates were determined 3 days post infection by detecting β-galactosidase activity in cellular lysates using the Tropix Gal-Screen kit (Applied Biosystems, Life Technologies, Frederick, MD) and the Orion microplate luminometer (Berthold, Bad Wildbad, Germany). All values represent reporter gene activities (relative light units per second; RLU/s) derived from triplicate infections minus background activities derived from uninfected cells.

To assess the effect of CLR01 and CLR03 on semen-mediated enhancement of HIV-1 infection, $10^4$ TZM-bl cells were seeded in 280 µl cell culture medium supplemented with 50 µg/ml gentamycin in 96-well flat-bottom plates the day before infection. Seminal plasma (20%) was treated with different concentrations of CLR01 or CLR03 (highest 925 µM) for 10 min at room temperature and then mixed with R5-tropic HIV-1 NL4-3 and CH058 (0.5 ng/ml p24 antigen). After 5 min, 20 µl of these mixtures were added to 280 µl TZM-bl cells. To minimize cytotoxic effects mediated by seminal plasma, the inoculums were replaced 2 hr later with fresh cell culture medium. Infection rates were determined as described above.

## Cell viability

The effect of CLR01, CLR03, nonoxynol-9, sodium dodecyl sulfate and Triton X-100 on the metabolic activity of TZM-bl cells was analyzed using the MTT assay. After 3 days of incubation, 20 µl of 5 mg/ml MTT (3-[4,5-dimethyl-2-thiazolyl]-2,5-diphenyl-2H-tetrazolium bromide) solution was added to the cells. After 4 hr the cell-free supernatant was discarded and formazan crystals were dissolved in 100 µl DMSO:Ethanol (1:2). Absorption was detected at 490 nm and corrected by the background absorption at 650 nm.

## Antiviral activity of CLR01 on HIV-1, HCMV, HSV-2, HCV, and adenovirus infection

### HIV-1

In the virus treatment assay, R5- and X4-tropic HIV-1 NL4-3, and HIV-1 transmitter/founder viruses CH058 and THRO were titrated with CLR01 or CLR03 (0–150 µM). After incubation for 10 min at 37°C, 20 µl of these mixtures were added to $10^4$ TZM-bl cells seeded 1 day prior in 180 µl cell culture medium. For the cell-treatment assay, DMEM instead of virus was titrated with CLR01 or CLR03, incubated, and added to TZM-bl cells analogous to the virus treatment protocol. After a 2-hr incubation at 37°C, old medium was replaced by fresh cell culture medium, and cells were infected with the different HIV-1 strains. β-galactosidase activity was measured 3 days post infection. $IC_{50}$ values were calculated with GraphPad Prism software.

### Human cytomegalovirus (HCMV)

Human foreskin fibroblasts (HFFs) were maintained in minimal essential medium (MEM, Invitrogen, Germany) supplemented with 10% fetal calf serum (Invitrogen), 2 mM L-glutamine (Biochrom AG, Germany), 100 U of penicillin and 100 µg of streptomycin (Gibco/BRL) per ml, and 1 × non-essential amino acids (Biochrom AG). Production of HCMV stock virus was a derivative of strain TB40-BAC4 (*Sinzger et al., 2008*), and all HCMV infection experiments were performed on HFF under serum-free conditions. To evaluate the inhibitory effect of compounds on virus entry, HCMV virus, corresponding to a multiplicity of infection of approximately 1 plaque forming unit, was pre-incubated with PBS, 100 µM CLR03, or 100 µM CLR01 for 30, 60, or 120 min, respectively, at 37°C in serum-free MEM. The virus/compound mixtures were then incubated for 16 hr with HFFs ($1.7 \times 10^4$/well) in a 96-well plate that was seeded 1 day prior to infection and washed twice with PBS before adding the mixtures. To determine infection rates, virus/compound mixtures were first removed by washing twice with PBS followed by fixation with ice-cold methanol for 10 min. HCMV-infected cells were visualized by indirect immunofluorescence staining for HCMV immediate-early (IE) antigen employing Mab13 (Argene) and cell nuclei staining by using 4′,6-diamidin-2-phenylindol (DAPI, Roche, Basel, Switzerland). HCMV infection rates for each compound were determined from images taken with the 10× objective lens and the fluorescence microscope Axio-Observer.Z1 (Zeiss, Germany) by counting numbers of IE-positive cells. Maximum inhibition of HCMV infection by CLR01 was found already after 30 min of incubation. The mean infection rate and standard deviation included the results from the different incubation times.

### Herpes simplex virus type 2 (HSV-2)

HSV-2 comprising a GFP reporter gene was treated with PBS, 100 µM CLR03, or 100 µM CLR01 and added to Vero cells. GFP-positive cells were counted using flow cytometry 2 days post infection.

### Hepatitis C virus (HCV)

HCV in vitro transcripts were generated and transfected using electroporation as described recently (*Koutsoudakis et al., 2006*). Harvested virus was then precipitated using PEG800 as described previously (*Lindenbach et al., 2005*) and resuspended in 10 mM HEPES, 150 mM NaCl. Huh-7.5 cells stably expressing firefly luciferase were seeded at a density of $2 \times 10^4$ cells per well of a 96-well plate 24 hr prior to inoculation. CLR01 and CLR03 were diluted in 10 mM HEPES, 150 mM NaCl and incubated with Renilla luciferase reporter virus particles (JcR-2a) (*Reiss et al., 2011*; *Haid et al., 2012*) for 5 min at 37°C. Cells were inoculated with JcR2-2a HCV in the presence of CLR01 or CLR03 for 72 hr at 37°C, washed with PBS, and lysed with 50 µl passive lysis buffer (Promega, Mannheim, Germany). To measure cytotoxicity, lysates were assayed for firefly luciferase activity using luciferin (200 µM luciferin, 25 mM glycylglycine, pH 8) in a plate luminometer (Lumat LB9507). For infectivity readout, lysates were assayed for Renilla luminescence using 1 µM coelenterazin (PJK, Kleinblittersdorf, Germany) in the same luminometer.

### Human adenovirus type 5 (HAdV5)

The E1-deleted replication-deficient human adenovirus type 5-based vector containing a HCMV promoter-controlled EGFP expression cassette was produced on N52.E6 cells (*Schiedner et al., 2000*), purified by one discontinuous and one continuous CsCl density gradient and subsequent size-exclusion chromatography (disposable PD-10, Amersham). The physical particle titer was determined by particle lysis and OD260 and confirmed by slot-blotting (*Kreppel et al., 2002*). To assess effects of CLR01 and CLR03 on the ability of the vector to transduce cells, the vector was titrated with 0–100 µM CLR01 or CLR03 and incubated 10 min at 37°C in 50 mM HEPES, 150 mM NaCl, pH 7.4. 1 day prior to infection, $10^5$ A549 cells per well were seeded in a 24-well format. Cells were infected with 200 MOI of the pretreated virus. EGFP expression was analyzed using a Beckman–Coulter Gallios flow cytometer 1 day post transduction.

## Impact of poly-L-lysine on the antiviral activity of CLR01

R5-tropic HIV-1 NL4-3 92TH014 was treated with 40 µM CLR01 in the presence of buffer or increasing concentrations of poly-L-lysine (molecular weight 70,000–150,000 by viscosity) (0.3–75 nM). After a 10-min incubation, TZM-bl cells were infected and β-galactosidase activity was measured 2 days post infection.

## Impact of coated BSA on the antiviral activity of CLR01

Microtiter plates (96 well, flat bottom, Sarstedt 83.3924) were coated over night with 100 µl of 0, 0.2, 1 or 5% BSA dissolved in PBS at 4°C. After washing three times with PBS the coated wells were filled with 70 µl of 900 µM CLR01 and incubated 30 min at 37°C. Then, CLR01 was serially diluted and added to HIV-1 NL4-3 92TH014. Following a 10-min incubation, TZM-bl cells were infected and β-galactosidase activity was measured 2 days post infection.

## Impact of free BSA on the antiviral activity of CLR01

R5-tropic HIV-1 NL4-3 92TH014 was treated with 220 µM CLR01 in the presence of buffer or increasing concentrations of BSA (0.01–10%). After a 10-min incubation, TZM-bl cells were infected and β-galactosidase activity was measured 2 days post infection.

## p24 release assay

HIV-1 NL4-3 92TH014 was incubated for 10 min at 37°C with PBS, 100 µM CLR03 or 100 µM CLR01 and centrifuged at 20,000×g and 4°C for 1 hr. The p24 content of the supernatant and pellet was determined using an in house p24-antigen ELISA.

## Atomic force microscopy of virions

Virus solutions (20 µl) were deposited on aminopropyl-modified glass cover slips (AP-Glass) and incubated for 1 hr at room temperature. After removing excess liquid, the deposited virus particles

were treated with 40 µl of 100 µM CLR01 or CLR03 and incubated for 10 min at RT. The samples were rinsed with PBS and imaged in PBS on a Nanowizard 3 AFM (JPK) in Quantitative Imaging mode using silicon nitride cantilevers with a spring constant of 0.03 N/m (Bruker, Billerica, MA).

## Giant unilamellar vesicles

Sodium dihydrogen phosphate, chloroform, and cholesterol (Chol) were purchased from Sigma–Aldrich. Lissamine rhodamine B 1,2-dihexadecanoyl-sn-glycero-3-phosphoethanolamine (*N*-Rh-DHPE), ATTO 647 were purchased from Life Technologies and ATTO-TEC, respectively. 1,2-dioleoyl-sn-glycero-3-phosphocholine (DOPC), sphingomyelin (SM), and 23-(dipyrrometheneboron difluoride)-24-norcholesterol (Bodipy-Chol) were purchased from Avanti Polar Lipids (Alabaster, AL). All chemicals used were of the highest analytical grade available and used without further purification.

Giant unilamellar vesicles (GUVs) were prepared by electroformation on optically transparent and electrically conductive indium tin oxide (ITO)-coated glass slides (SPI Supplies) in a preparation chamber consisting of a closed bath imaging chamber RC-21B affixed to a P-2 platform (both Warner Instruments Co.) topped with a flow-through temperature block. A solution of pure DOPC-containing 0.2 mol% *N*-Rh-DHPE or a lipid mixture of DOPC/SM/Chol (45/25/30 mol%) containing 0.2 mol% *N*-Rh-DHPE and 0.1 mol% Bodipy-Chol in chloroform was spread on an ITO-coated cover slip (20 µl, 1 mg/ml), spin-coated at 800 rpm for 1 min, and subsequently dried under vacuum for at least 2 hr. Afterwards, the lipids were hydrated in 10 mM $NaH_2PO_4$ pH 7.6, containing the water-soluble fluorophore ATTO 647 (5 µM) within the preparation chamber. The electroformation of pure DOPC and the DOPC/SM/cholesterol mixture was performed at room temperature and 60°C, respectively, by applying a frequency alternating current field (500 Hz, 100 mV for 10 min, 1 V for 20 min and 1.6 V for 2.5 hr) to the ITO electrodes by a function generator (Thurlby Thandar Instruments TG315). Afterwards, the preparation chamber was cooled down to room temperature in the case of the lipid mixture and carefully rinsed with 10 mM $NaH_2PO_4$ pH 7.6, to remove the water-soluble ATTO 647 that was not enclosed in the interior of the vesicles. Once a region of interest for imaging of the GUVs was chosen under the microscope, ~500 µl of CLR01 (150 µM) in 10 mM $NaH_2PO_4$ pH 7.6, was added. Images were recorded by a confocal laser-scanning microscope (Biorad MRC 1024) coupled via a side port to an inverted microscope (Nikon; Eclipse TE-300DV) enabling fluorescence excitation in the focal plane of an objective lens (Nikon Plan Apo 60× WI, NA 1.2). Fluorescence of Bodipy-Chol, *N*-Rh-DHPE, and ATTO 647 was sequentially acquired by alternating the excitation with the 488, 568, and 647 nm lines of a Kr/Ar laser (Dynamic Laser, Salt Lake City, UT, USA). Signals were detected in three different PMT channels (emission band pass filters 522 nm/FWHM [full width at half-maximum] 35 nm, 580 nm/FWHM 32 nm, and 680 nm/FWHM 32 nm). Image acquisition was controlled by the software LaserSharp2000 (Biorad). Analysis of the data was performed using the software Fiji (Max Planck Society for the Advancement of Science e.V., Munich, Germany). Images were background and brightness/contrast corrected.

## Atomic force microscopy of lipid bilayers

The phospholipids 1,2-dioleoyl-sn-glycero-3-phosphocholine (DOPC) and sphingomyelin (egg, chicken) (SM) were purchased from Avanti Polar Lipids (Alabaster, AL, USA). Sodium dihydrogen phosphate, chloroform, and cholesterol (Chol) were purchased from Sigma–Aldrich (Steinheim, BW, Germany). Stock solutions of 10 mg/ml lipids (DOPC, SM, Chol) in chloroform were dissolved to obtain 1.95 mg of total lipid with the composition of DOPC (100 mol%) and DOPC/SM/Chol (45/25/30 mol%) for the liquid AFM experiments. The majority of the chloroform was evaporated with a nitrogen stream and the rest of the solvent was removed afterwards by drying under vacuum overnight. The sodium dihydrogen phosphate buffer was filtered through filters of 0.02-µm pore size (Whatman, Dassel, Germany) before use. The dry lipid films were hydrated with 1 ml of 10 mM $NaH_2PO_4$, pH 7.6. Afterwards, the lipid mixtures were vortexed, kept in a water bath at 65°C for 15 min, and then sonicated for 10 min. After five freeze-thaw-vortex cycles and brief sonication, large multilamellar vesicles were formed and transformed to large unilamellar vesicles of uniform size by use of an extruder (Avanti Polar Lipids, Alabaster, USA) with polycarbonate membranes of 100 nm pore size at 65°C (*Weise et al., 2009, 2011*). The vesicle fusion on mica was carried out by depositing 35 µl of the extruded lipid vesicle solution together with 35 µl of $NaH_2PO_4$ buffer on freshly cleaved mica and incubation in a wet chamber at 70°C for 2 hr. After the vesicle fusion, the samples were rinsed

carefully with $NaH_2PO_4$ buffer to remove unspread vesicles. For the tweezer–membrane interaction studies, 800 µl of 150 µM CLR01 in $NaH_2PO_4$ buffer were slowly injected into the AFM fluid cell at room temperature and allowed to incubate for different time periods (1 min and 1 hr). Afterwards, the fluid cell was rinsed carefully with $NaH_2PO_4$ buffer before imaging to remove unbound tweezer. The measurements were performed on a MultiMode scanning probe microscope with a NanoScope IIIa controller (Bruker, Camarillo, CA, USA) and usage of a J-Scanner (scan size 125 µm). Images were obtained by applying the tapping mode in liquid with sharp nitride lever (SNL) probes mounted in a fluid cell (Bruker, Camarillo, CA, USA). Tips with nominal force constants of 0.24 N m$^{-1}$ were used at driving frequencies around 9 kHz and drive amplitudes between 343 and 570 mV. Slow scan frequencies between 1.0 kHz and 1.97 kHz were required for high-resolution images. The height and phase images of sample regions were acquired with resolutions of 512 × 512 pixels. All AFM measurements were carried out at room temperature and the partitioning of the tweezer was analyzed by using analysis and NanoScope version 5 processing software (*Weise et al., 2009*, *2011*).

## Acknowledgements

We thank Kelly Jordan-Sciutto, Mickey Marks, Mariana Torrente, Alice Ford, Elizabeth Sweeny, Korrie Mack, Annie Chen, Aaron Gitler, and Sandra Maday for helpful suggestions on the manuscript, Mary Leonard for artwork, and Dr Aleksey Lomakin for help with the DLS experiments. EL was funded by a fellowship of the Landesgraduiertenförderung Baden-Württemberg and is part of the International Graduate School in Molecular Medicine Ulm. Financial support was provided by grants from the Deutsche Forschungsgemeinschaft to JM and FK, and the Volkswagen Stiftung to CM, TW, FK, and JM. LMC was supported by an NSF Graduate Research Fellowship (DGE-0822). RMH was supported by supported by an HHMI grant awarded to Swarthmore College. JS was supported by an NIH Director's New Innovator Award (1DP2OD002177-01), a Bill and Melinda Gates Foundation Grand Challenges Explorations Award, and NIH grant R21HD074510. GB was supported by The UCLA Jim Easton Consortium for Alzheimer's Drug Discovery and Biomarker Development. TP was supported by a grant from the Helmholtz Association SO-024. ESG acknowledges a Liebig-stipend from the Fonds der Chemischen Industrie as well as the support of the Cluster of Excellence RESOLV (EXC 1069) and the Collaborative Research Center SFB1093, both funded by the Deutsche Forschungsgemeinschaft. GB, FGK, and TS are co-inventors of International Patent Application No. PCT/US2010/026419, USA Patent 8,791,092, European Patent Application 10 708 075.6.

## Additional information

### Competing interests

F-GK: Co-inventor of International Patent Application No. PCT/US2010/026419, USA Patent 8,791,092, European Patent Application 10 708 075.6 TS: Co-inventor of International Patent Application No. PCT/US2010/026419, USA Patent 8,791,092, European Patent Application 10 708 075.6 GB: Co-inventor of International Patent Application No. PCT/US2010/026419, USA Patent 8,791,092, European Patent Application 10 708 075.6 The other authors declare that no competing interests exist.

### Funding

| Funder | Grant reference | Author |
| --- | --- | --- |
| Deutsche Forschungsgemeinschaft (DFG) | | Frank-Gerrit Klärner, Elsa Sanchez-Garcia, Jan Münch |
| Volkswagen Foundation | | Christoph Meier, Frank Kirchhoff, Tanja Weil, Jan Münch |
| National Science Foundation (NSF) | Graduate Research Fellowship (DGE-0822) | Laura M Castellano |
| National Institutes of Health (NIH) | Director's New Innovator Award (DP2OD002177) | James Shorter |

| Funder | Grant reference | Author |
|--------|-----------------|--------|
| Bill and Melinda Gates Foundation | Grand Challenges Explorations Award | James Shorter |
| National Institute of Allergy and Infectious Diseases (NIAID) | R21HD074510 | James Shorter |
| University of California, Los Angeles | Jim Easton Consortium for Alzheimer's Drug Discovery | Gal Bitan |
| Fonds der Chemischen Industrie | | Elsa Sanchez-Garcia |
| Helmholtz Association | SO-024 | Thomas Pietschmann |
| Howard Hughes Medical Institute (HHMI) | Awarded to Swarthmore College | Rebecca M Hammond |

The funders had no role in study design, data collection and interpretation, or the decision to submit the work for publication.

### Author contributions

EL, LMC, RMH, KB-R, JS, Conception and design, Acquisition of data, Analysis and interpretation of data, Drafting or revising the article; CM, JS, NE, BS, CMS, SU, JE, GG, FK, TP, DP, OZ, Conception and design, Acquisition of data, Analysis and interpretation of data; VMH, DW, Conception and design, Drafting or revising the article, Contributed unpublished essential data or reagents; AS, BW, CH, Acquisition of data, Analysis and interpretation of data, Contributed unpublished essential data or reagents; FK, TW, F-GK, TS, ES-G, RW, JM, Conception and design, Analysis and interpretation of data, Drafting or revising the article; GB, Conception and design, Acquisition of data, Analysis and interpretation of data, Drafting or revising the article, Contributed unpublished essential data or reagents

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
