## [Decision Letter]

Thank you for sending your work entitled “A molecular tweezer antagonizes seminal amyloids and HIV infection” for consideration at *eLife*. Your article has been favorably evaluated by Randy Schekman (Senior editor) and three reviewers, one of whom, Jeffery W Kelly, is a member of our Board of Reviewing Editors.

The Reviewing editor and the other reviewers discussed their comments before we reached this decision, and the Reviewing editor has assembled the following comments to help you prepare a revised submission.

All reviewers concur that the significance of the work is potentially there. The concerns center around some of the data and critical control experiments relevant to the mechanism that seem to be missing.

All three reviewers expressed concerns centered around the mechanism of action of CRL01.

The authors postulate that classic host-guest complexes are forming whereas the reviewers suspect colloidal aggregation on the part of the CLR01 itself. Colloidal aggregation of organic small molecules can disrupt protein fibrils, by protein sequestration by the colloid (Feng et al., Nature Chem. Biol., 2008). Such a mechanism must be controlled for, especially given the peculiar efficacy of the compounds against the virion membranes themselves. Second, the authors should look to see if free lysine competes for CLR01 binding, disrupting its ability to act on the fibrils and the virions. There are few statistical analyses performed, so the relevance of comparing EC50 values is often unclear. On top of that, many of the raw data sets are of too low quality to produce accurate EC50 values, e.g. the lack of a saturation plateau. All three reviews are pasted below and the questions and suggestions in the reviews should be addressed if the authors choose to submit a revised manuscript.

*Reviewer #1*:

These results appear to be incredible, and they may be. There are several critical controls that appear to be lacking, as indicated below. If the data hold up after these controls, then I am enthusiastic. Amyloid fibrils in seminal fluid composed of at least 4 cationic proteins are established to enhance HIV transmission through intercourse by binding virons augmenting fusion efficiency. The Lys-binding molecular tweezer CLR01 delays the aggregation of PAP248-286 and SEM1(45-107) fibrillization, half-maximal inhibitory concentrations (IC50) of CLR01 inhibition exhibited were 2.6 μM and 18.8 μM, respectively. By contrast, the IC50 to impede PAP85-120 assembly was significantly higher at 970 μM. Have the authors controlled for the possibility that CLR01 is functioning through colloid formation? It is important that the authors show that CLR01 is not impeding the adsorption of the aggregates to the EM grid, which can distort what is in solution. If these reactions are run for more than 30h, does aggregation into fibrils eventually occur? Does seeded aggregation eventually occur? The critical control that CLR01 is not displacing ThT from the fibrils needs to be performed. The analog in which all eight lysine and arginine residues are replaced by alanine appears to be far from a perfect control in that this variant is anticipated to be much more amyloidogenic? In Figure 3, the critical control that CLR01 is not displacing ThT from the fibrils and inhibiting fibril adsorption to the grids needs to be run-the rapidity of disassembly is surprising, is it really disassembly? The result that CLR01 destroys retroviral particles with IC50 values between 10-20 μM by compromising virion integrity further suggests that CLR01 might be functioning as a colloid. CLR01 decreased viral infectivity of semen in a concentration-dependent manner, while CLR03 had no impact on semen-mediated infection enhancement. At a CLR01 concentration of 31 μM, the enhancing effect of semen was completely abolished. Thus, CLR01 prevents HIV infection in the presence of semen. If this is occuring through monomeric CLR01 this is incredible, but I suspect the authors' inhibitor is a micelle or colloid. The data are so impressive that I am concerned that these results are unlikely due to the monomeric molecular tweezer, but my hypothesis could be incorrect.

*Reviewer #2*:

Seminal amyloids are known to concentrate HIV virions and enhance infectivity. In this report, the authors use a recently identified, general inhibitor of amyloid formation (the molecular tweezer CLR01) to understand whether it also inhibits model seminal amyloids: SEVI, PAP85-120 and SEM1. CLR01 is known to work by docking to exposed lysine and arginine residues and this molecule has found use as a probe for biophysical chemistry and protein folding studies. Lump et al. first test CLR01 as a possible inhibitor of seminal amyloids in vitro, using EM and a ThT assay. They then test whether the molecules can reverse pre-formed amyloids using similar methods. Next, they test whether CLR01 might block binding of mock virions to fibrils and find that the compound seems to disrupt virions themselves. They use AFM and microscopy in vesicles and virus particles to test this idea – concluding that CLR01 seems to disrupt virion membranes, similar to a detergent. Somehow, this membrane disruption effect seems to be exclusive to viral membranes, but the mechanism remains cloudy (and somewhat confusing). They find that CLR01 blocks the infectivity-enhancing activity of seminal fibrils, both with and without addition of BSA. Based on these results, the possibility of using CLR01 for the prevention of viral spread are discussed. However, there are concerns (and even contradictions) about the lack of mechanistic knowledge and the nature of the CLR01-amyloid interaction is not well understood. The null hypotheses – that CLR01 is a non-specific lysine interactor or even a detergent – are not adequately tested. Without deeper mechanistic insight and more rigorous exploration of the binding events, publication of the work seems premature.

1) There are few statistical analyses performed, so the relevance of comparing EC50 values is often unclear. On top of that, many of the raw data sets are of too low quality to produce accurate values. For example, the value of 970 uM (Figure 2) has no saturation plateau, so one assumes that the value has a very large error. Similar problems can be seen in the raw data for Figure 2, Figure 2, Figure 3, Figure 3 and all of Figure 5. These results cannot be adequately fit because of sparse datapoints, so comparing (or even reporting) the half-maximal values is not advised.

2) One of the major problems with the manuscript is the uncertain mechanism of the compounds and the selectivity of the recognition. If CLR01 is simply blocking cationic charge of lysine and arginine residues, as the zeta potential and all-alanine mutant results imply, than one could presumably achieve this outcome faster, cheaper and more effectively with abundant biological polyanions, such as heparin or nucleic acids (DNA/RNA). What is the effect of these biological polyanions on the assembly of seminal amyloids and infectivity?

3) The hypothesis of CLR01 binding to individual lysine and arginine residues is not experimentally tested outside of a rather crude comparison of an all-alanine replacement peptide (which likely doesn't conform to the same structure as the native peptide). Does free lysine, arginine or other free organic cations (or even divalent cations) compete for CLR01 binding? Does systematic replacements of lysine residues would provide insight into what the key interactions might be? What is the affinity of these interactions? Presumably there are countless free lysine residues in true seminal fluid, so how can CLR01 have the selectivity required to identify the seminal amyloids (especially given the rather weak EC50 values in the mid-micromolar)?

4) The ability of the compounds to block fibril-induced virion activity in the presence of BSA doesn't appear to make logical sense. CLR01 was created as a general lysine/arginine probe – so why wouldn't the compound be adsorbed onto the surface of BSA, which carries surface-exposed nucleophiles? The affinity of CLR01 for the seminal amyloid peptides and BSA (as a stand-in for biological fluids) needs to be carefully performed – by NMR, ITC or other method.

5) The lack of CLR01-induced toxicity in mammalian cells is confusing, given the effects on vesicles and virion membranes (Figure 6). The mammalian cells have numerous cholesterol-rich, raft-like domains, so that argument doesn't work. Presumably, the mammalian membranes would be disrupted just like in the vesicle and virion studies, but not leading to toxicity. What is going on here? Is calcium flux activated? Do the cells change shape? What is the affinity of CLR01 for viral membranes, vesicles and mammalian membranes? What lipid is being bound? By what mechanism do the membranes dissolve? Normally, such questions could be addressed in subsequent work, but the readers of this document would be expected to have major concerns about the logical gas and contradictions in the datasets, making careful and unbiased discussion an important requirement.

6) What is the CMC equivalent of CLR01? In other words, at what concentration does it form micro-aggregates or micelles? Some (perhaps many?) of the observed activities could be a result of non-specific activities, as described by the Shoicet group. DLS studies and detergent competition experiments need to be performed.

*Reviewer #3*:

Lump and colleagues describe the activity of a host-guest cavitand-like molecule, CLR01, in disassembling seminal protein fibrils and directly disrupting the membrane of HIV (and related) virions, much reducing HIV virus infectivity. The cavitand-like CLR01 acts by binding to lysine or arginine amino acids (the guests), with previously measured affinities in the 20 μM range, and presumably this activity is what leads to de-fibrilization (although exactly why binding should lead to defibrilization is unclear; presumably binding is improved in the soluble form, though this is not stated in the manuscript). In principle this is an interesting story, and certainly a novel use of a cavitand-like molecule, one that might inspire other efforts to repurpose such molecules, which are well-known to chemists, for similar effects. The manuscript could have substantial impact. Before it is published, I suggest the authors address the following concerns:

1) My major concern regards the mechanism of action. The authors postulate that classic host-guest complexes are forming with the cationic amino acids, as shown in Figure 1, and this is important to the specificity and ultimate usefulness of the compounds. Currently, however, the manuscript lacks two controls that would help establish this mechanism:

A) First, the authors should control for colloidal aggregation on the part of the CLR01 itself. Colloidal aggregation of organic small molecules can disrupt protein fibrils, by protein sequestration in the small molecule colloid (Feng et al., Nature Chem. Biol., 2008). Such a mechanism must be controlled for, especially given the peculiar efficacy of the compounds against the virion membranes themselves (this is hard to square with a host-guest type interaction, but would make sense for colloidal aggregates). To do so, the authors should do one or more of the following i. Look to see if CLR01 is forming colloids in the 10 - 30 μM range, for instance by dynamic light scattering. ii. Add a mild, non-ionic detergent such as Tween-100 to disrupt the colloidal aggregates. This would right-shift the efficacy curves (assuming the detergent doesn't disrupt the fibrils directly). iii. Before addition to the fibrils, spin down the aqueous “solution” of CRL01 (not the DMSO one) in a microfuge at its highest speed for 20 minutes. Use the supernatant as the CLR01 solution for efficacy testing (colloids will typically pellet over this time). iv. Add 10 mg/ml BSA to the fibril mixture, before addition of CLR01. BSA will prophylacticly coat colloids, disrupting their actions.

B) Second, the authors should look to see if free lysine competes for CLR01 binding, disrupting its ability to act on the fibrils and the virions. If it does not, or the effect is small, this would cast doubt on the mechanism proposed by the authors. This to me is an obvious control, and I worry that it was in the manuscript and I just overlooked it – if I did I apologize. If not, it should be done.

Both controls should be done for both the effects on the protein fibrils and on the direct effects of the virus membrane. Whereas I hesitate to put authors to more work, I believe that these controls will substantially strengthen the manuscript, and should take more than a few days to perform.

2) Minor points:

A) I thought many of the dose response curves could be higher in quality, and should be re-done at better resolution.

B) In the third paragraph of the subsection “CLR01 disaggregates preformed SEVI and PAP85-120 fibrils” the authors write that CLR01 acts sub-stoichiometrically. How do they know what the concentration of the relevant fibril species is? What is the stoichiometry?

C) In the Discussion, the authors speculate on the use of CLR01 in vivo, as a drug or drug lead. Probably the only way such a molecule could be used is topically. It will not be orally absorbed (in all likelihood), and an intravenous route would lead to low compliance. If they are to speculate on this, as they do now, they should alert the readership to these points.

D) I was unconvinced by the Concentration-Response curves, which are crucial to the efficacy arguments for CLR01. For Figure 2, the curve should go through 0 (not 25%...how is this possible at no compound?). More observations are needed in the transition region, and the authors should certainly sample every half-log, not every log10 in concentration. For Figure 2, they should calculate a curve that actually goes through their points. I assume they fixed the Hill coefficient to 1, which doesn't seem to be supported by the curve. Here too, they need more points. In Figure 5, the Y-axis of the concentration-response curves should be on a linear scale, not a log scale. If it were, the reader would quickly notice that the curves are quite steep, which can be informative, and sometimes concerning.

Notwithstanding these concerns, the manuscript is interesting and could have wide impact. If the authors can address the concerns of the missing controls, I would support its publication.

---

## [Author Response]

*All reviewers concur that the significance of the work is potentially there. The concerns center around some of the data and critical control experiments relevant to the mechanism that seem to be missing*.

*All three reviewers expressed concerns centered around the mechanism of action of CRL01*.

*The authors postulate that classic host-guest complexes are forming whereas the reviewers suspect colloidal aggregation on the part of the CLR01 itself. Colloidal aggregation of organic small molecules can disrupt protein fibrils, by protein sequestration by the colloid (Feng et al., Nature Chem. Biol., 2008). Such a mechanism must be controlled for, especially given the peculiar efficacy of the compounds against the virion membranes themselves*.

We thank the editor and reviewers for raising this issue. We agree that it is critical to exclude the possibility that CLR01 is acting via a non-specific mechanism that involves colloidal CLR01 aggregates. Colloidal aggregates formed by certain organic small molecules can inhibit fibril assembly by various amyloidogenic proteins, but to the best of our knowledge have not been reported to disrupt preformed amyloid fibrils (also mentioned by reviewer 3) (14). Furthermore, in all our experiments performed with CLR01 during the past 8 years, no hint of colloidal aggregate formation has been found.

Here, to rule out any participation of colloidal aggregates, we performed five independent experiments to assess potential aggregate formation by CLR01 (documented and discussed in detail in the revised manuscript (subsection “CLR01 does not form colloidal aggregates”), new Figure 7). All experiments were performed in exactly the same buffer compositions as used in the manuscript:

1) NMR dilution titrations provided experimental evidence for a very weak CLR01 dimer formation (K_a_ = ∼60M^-1^) in PBS buffer (10 mM sodium phosphate, 137 mM NaCl, 2.7 mM KCl, pH 7.4), with ∼5% CLR01 dimers present at 500 µM (Figure 7; Figure 7—figure supplement 1). Dimerization was totally absent in HEPES buffer (25 mM HEPES, 150 mM KOAc, 10 mM Mg(OAc)_2_, pH 7.4) (Figure 7; Figure 7—figure supplement 2). Thus, CLR01 is predominantly monomeric.

2) Diffusion NMR (DOSY) experiments in both buffers revealed hydrodynamic radii of ∼0.9 to 1.0 nm, slightly above the monomeric species (Figure 7; Figure 7—figure supplement 3).

3) Microcalorimetric dilution titrations revealed only minute endothermic heat changes, strongly contradicting an extensive aggregation process (data not shown).

4) Pyrene fluorescence revealed that no critical micelle concentration (cmc) could be determined in a wide concentration range (0-500 µM CLR01) encompassing fibril assembly and remodeling conditions (Figure 7).

5) Dynamic Light Scattering (DLS), which overemphasizes the presence of large aggregates because the intensity of the scattered light is proportional to the square of the mass, revealed a very small fraction of CLR01 formed particles with hydrodynamic radius *R*_*H*_ ∼5–8 nm in PBS at concentration between 200–1,000 μM, plus a very minor fraction of particles with a *R*_*H*_ ∼ 20–40 nm (Figure 7). We did not observe CLR01 particles with an *R*_*H*_ of ∼95-400 nm, which is the size range typically associated with colloidal small-molecule aggregates (43). By contrast, no CLR01 particles could be detected at 10 or 50 μM in PBS (data not shown). Moreover, no CLR01 particles were detected in HEPES buffer up to 1 mM (Figure 7).

Thus, under all assay concentrations the vast majority of the tweezer molecules are monomeric. Minute amounts of dimers or higher order species may be present at high concentrations of CLR01 in some cases in PBS buffer (fibril assembly and cell culture buffer), but are absent from the HEPES buffer (fibril remodeling buffer). Thus, it is highly unlikely that colloidal CLR01 aggregates contribute meaningfully to the observed activity of the compound.

We have also performed experiments to exclude the possibility that the minute quantities of CLR01 that might form colloidal aggregates were the active species. Thus, the ability of CLR01 to: (a) inhibit fibril assembly by PAP248-286, PAP85-120, and SEM1(45-107); (b) remodel preformed SEVI and PAP85-120 fibrils; and (c) directly inhibit HIV infection (in the absence of seminal amyloid) was not affected by:

1) BSA (10-50mg/ml), which would adsorb to the surface of any colloidal CLR01 aggregates and quench their activity (Figure 2—figure supplement 4; Figure 3—figure supplement 2; Figure 5—figure supplement 2 and Figure 5—figure supplement 3; Figure 9) (14; 43).

2) Preclearing buffers containing CLR01 by centrifugation at 16,100-20,000g for 20min, which would remove any colloidal CLR01 aggregates (Figure 2—figure supplement 4; Figure 3—figure supplement 2; Figure 5—figure supplement 4) (44).

These data strongly suggest that the small quantities of colloidal CLR01 aggregates that might be present are not the active species.

Finally, a feature of small molecules that form colloidal aggregates is that they are non-specific inhibitors (14; 43; 44). By contrast, however, CLR01 displays specificity for amyloidogenesis by peptides that harbor lysine and arginine residues. Thus, if the lysine and arginine residues of PAP248-286 or PAP85-120 are replaced with alanine, then CLR01 can neither inhibit fibril assembly nor remodel preformed fibrils (Figures 2 and 3). This specificity is not consistent with an indiscriminate, non-specific mechanism of action mediated by colloidal CLR01 aggregates.

*Second, the authors should look to see if free lysine competes for CLR01 binding, disrupting its ability to act on the fibrils and the virions*.

We have now established that free lysine or poly-L-lysine disrupt the anti-amyloid and antiviral activities of CLR01 (Figure 2—figure supplement 3; Figure 3—figure supplement 1; Figure 4—figure supplement 1
Figure 5—figure supplement 1). These data provide further support that the lysine-binding cavity of CLR01 plays an important role in both the anti-amyloid and antiviral activities of CLR01.

*There are few statistical analyses performed, so the relevance of comparing EC50 values is often unclear. On top of that, many of the raw data sets are of too low quality to produce accurate EC50 values, e.g. the lack of a saturation plateau*.

We have now included additional replicates and tested additional concentrations of CLR01 to improve the accuracy of our IC_50_ and EC_50_ measurements (Figures 2 and 3). We now also qualify differences with statistics where appropriate throughout the manuscript (Figure 2—figure supplement 1, Figure 2—figure supplement 3, Figure 2—figure supplement 4 and Figure 2—figure supplement 5; Figure 3—figure supplement 1 and Figure 3—figure supplement 2; Figure 5—figure supplement 1, Figure 5—figure supplement 3 and Figure 5—figure supplement 4; Figure 6; Figure 8; Figure 9).

Reviewer #1:

*These results appear to be incredible, and they may be. There are several critical controls that appear to be lacking, as indicated below. If the data hold up after these controls, then I am enthusiastic. Amyloid fibrils in seminal fluid composed of at least 4 cationic proteins are established to enhance HIV transmission through intercourse by binding virons augmenting fusion efficiency. The Lys-binding molecular tweezer CLR01 delays the aggregation of PAP248-286 and SEM1(45-107) fibrillization, half-maximal inhibitory concentrations (IC50) of CLR01 inhibition exhibited were 2.6 μM and 18.8 μM, respectively. By contrast, the IC50 to impede PAP85-120 assembly was significantly higher at 970 μM. Have the authors controlled for the possibility that CLR01 is functioning through colloid formation*?

As outlined above in response to editor, we now provide evidence that CLR01 is not functioning through colloidal aggregate formation.

*It is important that the authors show that CLR01 is not impeding the adsorption of the aggregates to the EM grid, which can distort what is in solution*.

The reviewer raises a good point. When we mix CLR01 (or buffer) with preformed SEVI, PAP85-120, or SEM1(45-107) fibrils for 10 min (a time at which no fibril remodeling occurs; Figure 3) and then adsorb them to the EM grid, we observe abundant fibrils in both the CLR01 and buffer control conditions (Figure 2—figure supplement 2). Furthermore, it is evident in Figure 3, that CLR01 does not affect adsorption of SEM1(45-107) fibrils to the EM grid. Collectively, these observations strongly suggest that CLR01 does not impede adsorption of peptide conformers to the grid.

*If these reactions are run for more than 30h, does aggregation into fibrils eventually occur? Does seeded aggregation eventually occur*?

We have run the spontaneously fibril assembly reactions for 68 h for PAP248-286 (Figure 2) and 72 h for SEM1(45-107) (Figure 2) and observe very little fibril assembly in the presence of CLR01. We have now also run the PAP85-120 fibril assembly reaction for 68 h and observe very little fibril assembly in the presence of CLR01 (Figure 2). Thus, under these conditions CLR01 is a potent inhibitor, which prevents spontaneous PAP248-286, PAP85-120, and SEM1(45-107) fibril assembly.

*The critical control that CLR01 is not displacing ThT from the fibrils needs to be performed*.

We thank the reviewer for suggesting this key control. We have addressed this question in two ways. First, we have monitored early time points (0 min, 10 min and 20 min) in SEVI and PAP85-120 fibril remodeling by CLR01 (Figure 3). At these times, robust ThT fluorescence is still observed indicating that CLR01 is not simply displacing ThT from SEVI or PAP85-120 fibrils. Indeed, fibrils are still prevalent at these early times (Figure 2—figure supplement 2). Thus, CLR01 is not simply competing for ThT binding as we would expect an instantaneous effect on ThT fluorescence as is observed, for example, with another small molecule EGCG and ABeta40 fibrils (Palhano et al., 2013). In this regard, it is also important to note that ThT fluorescence of SEM1(45-107), SEVI(Ala), and PAP85-120(Ala) fibrils is unaffected by CLR01 at various times (Figure 3). Thus, CLR01 is not a general inhibitor or competitor of ThT binding to amyloid fibrils.

In a second approach, we employed a ThT displacement assay (36). Here, preformed SEVI, PAP85-120, or SEM1(45-107) fibrils (5 µM monomer) were preincubated with ThT (25 µM) for 30min at room temperature. Buffer, CLR01 (250 µM), or a known competitor of ThT binding, BTA-1 (250 µM) (36) were then added and incubated for 10min at room temperature. ThT displacement was then assessed by fluorescence. BTA-1 effectively displaced ThT, whereas CLR01 did not (Figure 2—figure supplement 1). These findings establish that CLR01 does not simply displace ThT from SEVI, PAP85-120, or SEM1(45-107) fibrils. Thus, any reduction in ThT fluorescence caused by CLR01 can be attributed to an inhibition of fibril assembly or remodeling of fibrils into conformers unable to bind ThT. Inhibition of fibril assembly by CLR01 has also been assessed by EM (Figure 2). Likewise, fibril remodeling by CLR01 has also been assessed by: EM (Figure 3), Proteostat staining, a red fluorescent aggregate sensing dye (Figure 3), seeding activity of remodeled conformers (Figure 3), and circular dichroism (Figure 3). By employing multiple approaches, we thus establish that CLR01 inhibits amyloid assembly of PAP248-286, PAP85-120, and SEM1(45-107), and remodels preformed SEVI and PAP85-120 fibrils.

*The analog in which all eight lysine and arginine residues are replaced by alanine appears to be far from a perfect control in that this variant is anticipated to be much more amyloidogenic*?

We have demonstrated that PAP248-286(Ala) and now the PAP85-120(Ala) peptide readily assemble into ThT-reactive fibrils in the presence of CLR01 (Figure 2). Moreover, SEVI(Ala) and PAP85-120(Ala) fibrils are resistant to remodeling by CLR01 (Figure 3). Thus, the activity of CLR01 cannot be attributed to a general ability to displace ThT from fibrils (and see above response). Indeed, CLR01 also has no effect on the ThT fluorescence of preformed SEM1(45-107) fibrils (Figure 3). Rather, the PAP248-286(Ala) and PAP85-120(Ala) results suggest that CLR01 is unable to inhibit fibrillization or remodel fibrils when the target polypeptide is devoid of lysine and arginine residues. The PAP248-286(Ala) and PAP85-120(Ala) peptides spontaneously assemble into amyloid more rapidly than the wild-type peptides (Figure 2). However, we have been unable to establish conditions (e.g. higher CLR01 concentrations) where CLR01 prevents assembly of PAP248-286(Ala) and PAP85-120(Ala) into amyloid fibrils. Likewise, we have been unable to establish conditions (e.g. higher CLR01 concentrations) where CLR01 remodels SEVI(Ala) or PAP85-120(Ala) fibrils. These results are consistent with our suggestion that CLR01 is inactive against polypeptides devoid of lysine or arginine residues.

*In*
Figure 3*, the critical control that CLR01 is not displacing ThT from the fibrils and inhibiting fibril adsorption to the grids needs to be run-the rapidity of disassembly is surprising, is it really disassembly*?

As discussed above, we have demonstrated that CLR01 does not affect amyloid adsorption to EM grids (Figure 2—figure supplement 2) or displace ThT from fibrils (Figure 2—figure supplement 1). Although rapid, we now show that fibril remodeling by CLR01 is not instantaneous. Indeed, SEVI and PAP85-120 fibrils resist remodeling for ∼20 min (Figure 3). However, to be more cautious we now avoid the term fibril disassembly, and instead use the term fibril remodeling, which more accurately describes the observed effects.

*The result that CLR01 destroys retroviral particles with IC50 values between 10-20 μM by compromising virion integrity further suggests that CLR01 might be functioning as a colloid. CLR01 decreased viral infectivity of semen in a concentration-dependent manner, while CLR03 had no impact on semen-mediated infection enhancement. At a CLR01 concentration of 31 μM, the enhancing effect of semen was completely abolished. Thus, CLR01 prevents HIV infection in the presence of semen. If this is occuring through monomeric CLR01 this is incredible, but I suspect the authors' inhibitor is a micelle or colloid. The data are so impressive that I am concerned that these results are unlikely due to the monomeric molecular tweezer, but my hypothesis could be incorrect*.

As outlined in the response to the editor, CLR01 does not form colloids, micelles, or other large aggregates (Figure 7). We agree that the µM activity of CLR01 against HIV-1 in the presence of semen is remarkable. However, this is not due to formation of colloidal CLR01 aggregates as human seminal fluid contains abundant quantities of protein, which would be anticipated to quench the activity of a non-specific colloidal aggregate by adsorption (14). Moreover, CLR01 activity against the virus is preserved in the presence of BSA (10 mg/ml; Figure 5—figure supplement 2 and Figure 5—figure supplement 3) and is retained in the soluble fraction after centrifugation of CLR01 (Figure 5—figure supplement 4), two established methods to eliminate the activity of non-specific small molecule colloids (14). Furthermore, CLR01 exerts powerful inhibitory effects on amyloid accumulation of Abeta and TTR in vivo in mouse models (5; 15) and alpha-synuclein aggregation in zebrafish models (64). This potent activity in vivo makes a colloidal-based mechanism extremely unlikely. Taken together, these data suggest that CLR01 is not acting via a mechanism involving non-specific colloidal CLR01 aggregates.

Reviewer #2:

*[… The authors] find that CLR01 blocks the infectivity-enhancing activity of seminal fibrils, both with and without addition of BSA. Based on these results, the possibility of using CLR01 for the prevention of viral spread are discussed. However, there are concerns (and even contradictions) about the lack of mechanistic knowledge and the nature of the CLR01-amyloid interaction is not well understood. The null hypotheses – that CLR01 is a non-specific lysine interactor or even a detergent – are not adequately tested. Without deeper mechanistic insight and more rigorous exploration of the binding events, publication of the work seems premature*.

We agree that further mechanistic insight would enhance our study. Hence, we have now added several experiments that provide deeper insight into the mechanism of CLR01 action. We can now exclude the possibility that CLR01 is acting via a non-specific mechanism mediated by colloidal CLR01 aggregates or detergent-like CLR01 micelles (see response to editor). We also demonstrate that the lysine-binding cavity of CLR01 plays an important role in both the anti-amyloid and antiviral activities of CLR01 (Figure 2—figure supplement 3; Figure 3—figure supplement 1; Figure 4—figure supplement 1; Figure 5—figure supplement 1). Indeed, it is important to note that the specificity of CLR01 binding to lysine or arginine is well established. CLR01 was tested against almost all naturally occurring amino acid residues in model compounds, and does not bind to any other amino acid side chain except for lysine (*K*_*d*_∼10 µM) and arginine (*K*_*d*_∼50 µM) (16; 29; 31). The structure of the CLR01-lysine complex and the precise mechanism of lysine threading into the CLR01 guest cavity and subsequent ion pairing have been extensively characterized by NMR spectroscopy, crystal structure, molecular dynamics, and quantum mechanics/molecular mechanics (QM/MM) calculations (6; 13; 30). Importantly, CLR01 has no effect on formation or remodeling of SEVI(Ala) or PAP85-120(Ala) fibrils in which all lysine and arginines residues are exchanged to alanines (Figures 2 and 3). Thus, CLR01 does not inhibit amyloid formation or remodel amyloid fibrils in a non-specific way but rather requires that the polypeptides bear arginine or lysine residues. Moreover, the presence of arginine or lysine residues in the target peptide also does not guarantee that CLR01 will be able to remodel amyloid fibrils. Indeed, CLR01 could not remodel preformed SEM1(45-107) fibrils (Figure 3), indicating a specific effect of CLR01 on SEVI and PAP85-120 fibrils (Figure 3). Importantly, we now also show that free lysine or poly-L-lysine abrogate the anti-amyloid activity of CLR01, further demonstrating specificity (Figure 2—figure supplement 3; Figure 3—figure supplement 1; Figure 4—figure supplement 1). Moreover, CLR01 does not exert cytotoxic effects on cells, unlike non-ionic and anionic detergents such as TX-100, nonoxyl 9, or SDS (Figure 4—figure supplement 3). Collectively, these data establish that CLR01 is not acting in a non-specific manner that involves detergent-like CLR01 micelles.

*1) There are few statistical analyses performed, so the relevance of comparing EC50 values is often unclear. On top of that, many of the raw data sets are of too low quality to produce accurate values. For example, the value of 970 uM (*Figure 2*) has no saturation plateau, so one assumes that the value has a very large error. Similar problems can be seen in the raw data for*
Figure 2*,*
Figure 2*,*
Figure 3*,*
Figure 3
*and all of*
Figure 5*. These results cannot be adequately fit because of sparse datapoints, so comparing (or even reporting) the half-maximal values is not advised*.

We have now included additional replicates and tested additional concentrations of CLR01 to improve the accuracy of our IC_50_ and EC_50_ measurements (Figures 2 and 3). We now also qualify differences with statistics where appropriate throughout the manuscript (Figure 2—figure supplement 1, Figure 2—figure supplement 3, Figure 2—figure supplement 4 and Figure 2—figure supplement 5; Figure 3—figure supplement 1 and Figure 3—figure supplement 2; Figure 5—figure supplement 1, Figure 5—figure supplement 3 and Figure 5—figure supplement 4; Figure 6; Figure 8; Figure 9).

The data shown in Figure 5 were derived from triplicate infections with seven different concentrations of the tweezer including controls (infected and uninfected cells). The inhibition curve follows a typical sigmoidal shape and thus allows a meaningful determination of IC_50_ values using Graphpad Prism software_._ We have extensive experience in the determination of antiviral activities and IC_50_ values, including the preclinical and clinical development of drugs (Forssmann et al., 2010; [51]; Venken et al., 2011; [96]; Zirafi et al., 2015).

*2) One of the major problems with the manuscript is the uncertain mechanism of the compounds and the selectivity of the recognition. If CLR01 is simply blocking cationic charge of lysine and arginine residues, as the zeta potential and all-alanine mutant results imply, than one could presumably achieve this outcome faster, cheaper and more effectively with abundant biological polyanions, such as heparin or nucleic acids (DNA/RNA). What is the effect of these biological polyanions on the assembly of seminal amyloids and infectivity*?

The reviewer raises an interesting point. However, it is important to note that CLR01 and CLR03 have identical negative charge, but only CLR01 exerts an anti-amyloid or antiviral effect (Figures 2, 3, 5, 6, 8 and 9). We can therefore exclude the possibility that the effects of CLR01 are a trivial consequence of being anionic. Simply being anionic is not sufficient for anti-amyloid or antiviral activity, as CLR03 lacks activity (Figures 2, 3, 5, 6, 8 and 9). This can be rationalized by the fact that, at the molecular level, the interaction of CLR01 with Lys residues is based on the formation of inclusion complexes in which Lys is trapped inside the tweezer’s cavity (6; 13). The chemical structure of CLR03 does not allow for the formation of such inclusion complexes. Thus, as shown by the simulations reported in this paper, CLR03 interactions with the Lys residues of PAP are not preserved, unlike CLR01, which forms conserved inclusion complexes (Figure 1).

With regard to the effect of biological polyanions on HIV infectivity, it is well established that heparin (Ito et al., 1987; Rider, 1997; Vives et al., 2005) or oligodeoxynucleotides (Matsukura et al., 1987; Stein et al., 1993; Vaillant et al., 2006) can inhibit HIV-1 infection by binding to viral envelope glycoproteins, cellular receptors, or both. By contrast, CLR01 directly disrupts the viral membrane thereby eliminating the virus (Figure 6). Hence, CLR01 and polyanions directly inhibit HIV infection by very different mechanisms.

It has also been demonstrated that heparin binds SEVI or PAP85-120 fibrils by electrostatic interactions, rendering the fibril surface negatively charged, thereby preventing enhancement of HIV infection (3; 69). Even though CLR01 is also negatively charged, it binds to SEVI, PAP85-120, or SEM1(49-107) fibrils via specific interaction with lysine or arginine residues making the fibrils neutrally charged (Figure 4). In addition, CLR01 prevents SEVI, PAP85-120, or SEM1(45-107) fibril formation (Figure 2) and remodels preformed SEVI and PAP85-120 fibrils (Figure 3), whereas polyanions such as heparin accelerate the formation of SEVI fibrils (Tan et al., 2013). Thus, CLR01 and polyanions inhibit the viral infection enhancing activity of amyloids fibrils by different mechanisms.

Polyanions, including heparin and various other glycosaminoglycans, have been investigated as HIV microbicides for decades, as they are also thought to exert direct anti-HIV activity by inhibiting the binding of the viral envelope glycoprotein, gp120, to its major cell surface receptor CD4 (Callahan et al., 1991; Rider, 1997; Witvrouw and De Clercq, 1997). Unfortunately, such anionic polymers have proven to be unsuccessful in past clinical trials due to their poor bioavailability and induction of inflammatory responses in the genital tract, which instead augment HIV transmission by recruiting HIV-susceptible target cells to the genital mucosa (40; 83). Moreover, it has recently become clear that the HIV-enhancing activity of semen diminishes the efficacy of diverse anti-HIV microbicides, including several polyanions (96). By contrast, CLR01 retains potent antiviral activity in the presence of semen (Figure 9). Thus, CLR01 antagonizes seminal amyloid and HIV via distinct mechanisms to polyanions and retains high activity in semen, which is likely an advantageous property for a potential microbicide candidate. We now discuss some of these issues in the Discussion.

*3) The hypothesis of CLR01 binding to individual lysine and arginine residues is not experimentally tested outside of a rather crude comparison of an all-alanine replacement peptide (which likely doesn't conform to the same structure as the native peptide)*.

We have established that CLR01 is ineffective in preventing amyloid formation by PAP248-286(Ala) (Figure 2), and is also unable to remodel preformed SEVI(Ala) fibrils (Figure 3). We have now extended these experiments to a PAP85-120 peptide, PAP85-120(Ala), where all the lysines and arginines are replaced by alanine. CLR01 is unable to inhibit the amyloidogenesis of PAP85-120(Ala) (Figure 2) and is unable to remodel preformed PAP85-120(Ala) fibrils (Figure 3). These findings strongly suggest that lysine and arginine residues are critical for CLR01 activity against PAP248-286 and PAP85-120.

It is important to note that it is well established that CLR01 selectively binds lysine in amyloid fibrils (1; 5; 15; 64; 74; 75; 95). We have examined several complexes between CLR01 and peptides/proteins with multiple lysines (16; 29; 31). 2D NMR and MS experiments have revealed, for example, that the tweezer molecule sits exclusively on both lysines and to a smaller extent on the single arginine of Αβ1-40 (75). In addition, we have shown that some enzymes with lysine residues around the active site can be inhibited by CLR01, and start to work again, after addition of free lysine (78). As mentioned above, the interaction between CLR01 and lysine has been studied and it is structurally well characterized: threading of the lysine side chain through the tweezer cavity, plus ion pair formation between the lysine ammonium cation and the tweezer phosphate anion produces a *K*_*d*_ of ∼10 µM for the 1:1 complex between one lysine side chain and one tweezer molecule, as determined by NMR and fluorescence titration (6; 30; 37). Collectively, these precedents provide further support for a critical role for CLR01 interactions with lysine or arginine residues in the effects we observe on seminal amyloid peptides.

*Does free lysine, arginine or other free organic cations (or even divalent cations) compete for CLR01 binding*?

To provide evidence that the anti-amyloid activity of CLR01 depends on the specific interaction with lysine, we performed competition experiments with free lysine or poly-L-lysine. As now shown in the revised manuscript, pretreatment of CLR01 with free lysine or poly-L-lysine abrogates the tweezers’ ability to: block amyloid formation (Figure 2—figure supplement 3), remodel amyloid (Figure 3—figure supplement 1), neutralize amyloid surface charge (Figure 4—figure supplement 1), and directly antagonize viral infection (Figure 5—figure supplement 1).

*Does systematic replacements of lysine residues would provide insight into what the key interactions might be? What is the affinity of these interactions*?

The reviewer makes great suggestions here. It will be extremely interesting to systematically replace the various lysine and arginine residues in PAP248-286, PAP85-120, and SEM1(45-107) to determine which lysines or arginines are critical for CLR01 to prevent amyloid assembly by these peptides. Likewise, it will be very interesting to systematically replace the various lysine and arginine residues in PAP248-286 and PAP85-120 to determine which are critical for fibril remodeling by CLR01. How these alterations affect the affinity of CLR01 for the various peptides will also be very interesting to determine. However, this is a very large set of studies, which we suggest is beyond the scope of this initial manuscript. We plan to perform these for a separate study where we dissect CLR01 mechanism in further detail.

*Presumably there are countless free lysine residues in true seminal fluid, so how can CLR01 have the selectivity required to identify the seminal amyloids (especially given the rather weak EC50 values in the mid-micromolar)*?

The reviewer raises a very interesting point. The interaction between a single lysine residue and CLR01 has a *K*_*d*_ of ∼10µM, with fast on and off rates. On proteins, however, only a few lysines are well accessible for the relatively large molecular skeleton of the tweezers. For example, ITC and MD+QM/MM calculations have revealed that only 5 of 17 surface lysines in a 14-3-3 protein are complexed by CLR01 (6). In addition, a key point is that, because of the high on–off rate, CLR01 disrupts only relatively weak interactions as well as events that depend on critical and accessible lysine or arginine residues. Therefore, the tweezer has shown selectivity in multiple cases that on first impression may appear counter-intuitive. For example, inhibition of abnormal protein aggregation is achieved at 1:1–1:5 protein:CLR01 concentration (1; 74; 75), whereas disruption of controlled protein polymerization, e.g., of tubulin, requires 55-fold excess CLR01 (4) and enzyme inhibition requires ∼1,000-fold excess CLR01 (78). Moreover, in vivo, CLR01 shows no toxicity at doses 2–3 orders of magnitude above those needed to enable clearance of amyloid (4; 5; 15; 64). Thus, selectivity for lysine-bearing amyloid conformers in complex biological fluids is achieved by CLR01. We now discuss the issue (Discussion).

To experimentally address this question, we have tested the effect of BSA on the ability of the tweezer to antagonize virus infection. Neither coated nor free BSA affected the antiviral activity of CLR01 (Figure 5—figure supplement 2 and Figure 5—figure supplement 3). Of note, all cell culture experiments (Figures 4, 5, 8 and 9) were performed in the presence of 10% FCS which contains abundant proteins, further demonstrating that free lysines at the surface of bulk, structured proteins do not affect CLR01 function. Likewise, CLR01 retains antiviral activity in the presence of seminal fluid (Figure 9), which also contains abundant proteins bearing lysine residues. Moreover, inhibition of amyloid assembly or amyloid remodeling by CLR01 was unaffected by BSA, indicating that surface lysines and arginines on BSA are not effective competitors of CLR01 activity (Figure 2—figure supplement 4; Figure 3—figure supplement 2).

*4) The ability of the compounds to block fibril-induced virion activity in the presence of BSA doesn't appear to make logical sense. CLR01 was created as a general lysine/arginine probe – so why wouldn't the compound be adsorbed onto the surface of BSA, which carries surface-exposed nucleophiles? The affinity of CLR01 for the seminal amyloid peptides and BSA (as a stand-in for biological fluids) needs to be carefully performed – by NMR, ITC or other method*.

There is a profound difference between freely accessible lysines on non-structured protein domains, which can all be complexed by the tweezers, and folded proteins, whose surface often does not allow the sterically-demanding tweezer molecule to approach a lysine residue close enough for the threading mechanism. The interaction of CLR01 with seminal fibrils has been discussed above. To experimentally address whether BSA affects the antiviral activity of the tweezer and its ability to antagonize fibril-mediated infectivity enhancement (the two main mechanisms to prevent sexual viral transmission) we performed competition experiments. As mentioned above, we found that physiologically relevant concentrations of coated or free BSA do not affect the direct antiviral activity of CLR01 (Figure 5—figure supplement 2 and Figure 5—figure supplement 3). These results are in agreement with our previous data showing that the tweezer is active in the presence of semen (Figure 9). BSA also failed to antagonize inhibition of amyloid assembly or amyloid remodeling by CLR01, suggesting that surface lysines and arginines on BSA are ineffective competitors of CLR01 (Figure 2—figure supplement 4; Figure 3—figure supplement 2).

*5) The lack of CLR01-induced toxicity in mammalian cells is confusing, given the effects on vesicles and virion membranes (*Figure 6*). The mammalian cells have numerous cholesterol-rich, raft-like domains, so that argument doesn't work. Presumably, the mammalian membranes would be disrupted just like in the vesicle and virion studies, but not leading to toxicity. What is going on here? Is calcium flux activated? Do the cells change shape? What is the affinity of CLR01 for viral membranes, vesicles and mammalian membranes? What lipid is being bound? By what mechanism do the membranes dissolve? Normally, such questions could be addressed in subsequent work, but the readers of this document would be expected to have major concerns about the logical gas and contradictions in the datasets, making careful and unbiased discussion an important requirement*.

As mentioned by the reviewer, we suggest that a thorough analysis of the interaction of the tweezer with cellular and viral membranes is beyond the scope of this study. We like to point out, however, that we have extensively studied possible cytotoxic effects of CLR01 in vitro and in vivo, but have never observed significant effects at concentrations up to 500 µM (Figure 4—figure supplement 3). We now also show in new Figure 4—figure supplement 3, that CLR01 is far less cytotoxic than various detergents, including Triton X-100, SDS, or N9. The tweezer also has been tested *in vivo* in multiple experiments, as mentioned above, and was found to have a high safety margin in mice (4). As recommended, we now discuss possible mechanisms of membrane disruption in the revised Discussion. Finally, it is important to note that the HIV membrane possesses a highly unusual lipid composition, which is distinct even from the detergent-resistant microdomains typically found in the plasma membrane, indicating that HIV budding involves a specific lipid raft clustering process (8). For example, although the HIV membrane resembles lipid raft microdomains in that it is enriched for saturated lipids, phosphatidylserine, plasmalogen-phosphatidylethanolamine, cholesterol, and sphingolipids, it is unusually enriched for the unusual sphingolipid dihydrosphingomyelin (8; 38). This unusual lipid raft composition could make the viral membrane uniquely susceptible to the action of CLR01. In our future work, we intend to address this interesting point in greater detail.

*6) What is the CMC equivalent of CLR01? In other words, at what concentration does it form micro-aggregates or micelles? Some (perhaps many?) of the observed activities could be a result of non-specific activities, as described by the Shoicet group. DLS studies and detergent competition experiments need to be performed*.

As outlined in detail in the response to the editor, we performed a series of experiments to exclude that CLR01 forms detergent-like micelles or colloidal aggregates. Thus, we show that even at millimolar concentrations the tweezers are present as predominantly monomeric species; only in PBS buffer does a very weak dimerization occur and minute amounts of higher-order species can be observed by DLS, due to the strong bias of this technique for large particles. No cmc could be determined using pyrene fluorescence. These experiments are summarized in the new Figure 7. We have also performed a series of experiments to exclude the possibility that the minute quantities of higher order CLR01 structures are the active species, i.e.: CLR01 activity is not affected by BSA (which would adsorb to the colloid and quench activity) or by preclearing any colloidal aggregates by centrifugation (Figure 2—figure supplement 4; Figure 3—figure supplement 2; Figure 5—figure supplement 2, Figure 5—figure supplement 3 and Figure 5—figure supplement 4). Likewise, CLR01 is highly active in the presence of FCS (Figures 4 and 8) and semen (Figure 9), complex multiprotein mixtures that would also be predicted to quench the non-specific activity of colloidal CLR01 aggregates. Unfortunately, we were unable to perform the suggested experiments with non-ionic detergents as these exerted inhibitory effects on amyloidogenesis by seminal peptides, which conflated analysis. A similar issue of detergent incompatibility with amyloid assembly was encountered by Feng et al. (14).

Reviewer #3:

*[…] The manuscript could have substantial impact. Before it is published, I suggest the authors address the following concerns*:

*1) My major concern regards the mechanism of action. The authors postulate that classic host-guest complexes are forming with the cationic amino acids, as shown in*
Figure 1*, and this is important to the specificity and ultimate usefulness of the compounds. Currently, however, the manuscript lacks two controls that would help establish this mechanism*:

*A) First, the authors should control for colloidal aggregation on the part of the CLR01 itself. Colloidal aggregation of organic small molecules can disrupt protein fibrils, by protein sequestration in the small molecule colloid (Feng et al., Nature Chem. Biol., 2008). Such a mechanism must be controlled for, especially given the peculiar efficacy of the compounds against the virion membranes themselves (this is hard to square with a host-guest type interaction, but would make sense for colloidal aggregates). To do so, the authors should do one or more of the following i. Look to see if CLR01 is forming colloids in the 10 - 30 μM range, for instance by dynamic light scattering. ii. Add a mild, non-ionic detergent such as Tween-100 to disrupt the colloidal aggregates. This would right-shift the efficacy curves (assuming the detergent doesn't disrupt the fibrils directly). iii. Before addition to the fibrils, spin down the aqueous “solution” of CRL01 (not the DMSO one) in a microfuge at its highest speed for 20 minutes. Use the supernatant as the CLR01 solution for efficacy testing (colloids will typically pellet over this time). iv. Add 10 mg/ml BSA to the fibril mixture, before addition of CLR01. BSA will prophylacticly coat colloids, disrupting their actions*.

As outlined above in the response to editor, we now provide evidence that CLR01 is not functioning through colloidal aggregate formation. Thus, CLR01 is predominantly monomeric and only forms minute quantities of colloidal aggregates in PBS buffer, and does not form them at all in HEPES buffer (Figure 7). Even then, CLR01 activity is unaffected by BSA or by preclearing aqueous solutions of CLR01 (incidentally, stock CLR01 solutions were made in water or PBS and not DMSO, p. 23) via centrifugation (Figure 2—figure supplement 4; Figure 3—figure supplement 2; Figure 5—figure supplement 2, Figure 5—figure supplement 3 and Figure 5—figure supplement 4), indicating that even if present, colloidal CLR01 particles cannot be the active species. Unfortunately, we were unable to perform the suggested experiments with non-ionic detergents as these exerted inhibitory effects on amyloidogenesis by seminal peptides, which conflated analysis. A similar issue of detergent incompatibility with amyloid assembly was encountered by Feng et al. (14).

*B) Second, the authors should look to see if free lysine competes for CLR01 binding, disrupting its ability to act on the fibrils and the virions. If it does not, or the effect is small, this would cast doubt on the mechanism proposed by the authors. This to me is an obvious control, and I worry that it was in the manuscript and I just overlooked it – if I did I apologize. If not, it should be done*.

*Both controls should be done for both the effects on the protein fibrils and on the direct effects of the virus membrane. Whereas I hesitate to put authors to more work, I believe that these controls will substantially strengthen the manuscript, and should take more than a few days to perform*.

As outlined above in the response to the editor, we performed these important controls. As expected from all previously published data on CLR01, we found that free lysine and poly-L-lysine abrogate the anti-amyloid and antiviral effects of CLR01 (Figure 2—figure supplement 3; Figure 3—figure supplement 1; Figure 4—figure supplement 1; Figure 5—figure supplement 1).

*2) Minor points*:

*A) I thought many of the dose response curves could be higher in quality, and should be re-done at better resolution*.

We have now included additional replicates and tested additional concentrations of CLR01 to improve the accuracy of our IC_50_ and EC_50_ measurements (Figures 2 and 3).

*B) In the third paragraph of the subsection “CLR01 disaggregates preformed SEVI and PAP85-120 fibrils” the authors write that CLR01 acts sub-stoichiometrically. How do they know what the concentration of the relevant fibril species is? What is the stoichiometry*?

By substoichiometric we meant relative to peptide monomers not fibrils. We now clarify this point in the manuscript (subsection “CLR01 remodels preformed SEVI and PAP85-120 fibrils”).

*C) In the Discussion, the authors speculate on the use of CLR01 in vivo, as a drug or drug lead. Probably the only way such a molecule could be used is topically. It will not be orally absorbed (in all likelihood), and an intravenous route would lead to low compliance. If they are to speculate on this, as they do now, they should alert the readership to these points*.

Of course a topical application of CLR01 as microbicide is the only possible application, we hope that we have clarified this point in the revised manuscript (see first paragraph in the Discussion).

*D) I was unconvinced by the Concentration-Response curves, which are crucial to the efficacy arguments for CLR01. For*
Figure 2*, the curve should go through 0 (not 25%...how is this possible at no compound?). More observations are needed in the transition region, and the authors should certainly sample every half-log, not every log10 in concentration*.

We have now adjusted our fits so the curves go through zero. We have now included additional replicates and tested additional concentrations of CLR01 to improve the accuracy of our IC_50_ and EC_50_ measurements (Figures 2 and 3).

*For*
Figure 2*, they should calculate a curve that actually goes through their points. I assume they fixed the Hill coefficient to 1, which doesn't seem to be supported by the curve. Here too, they need more points*.

The reviewer is correct, we had fixed the Hill coefficient to 1. We now fit with a variable slope and have included additional replicates and tested additional concentrations of CLR01 to improve the accuracy of our IC_50_ measurement in Figure 2.

*In*
Figure 5*, the Y-axis of the concentration-response curves should be on a linear scale, not a log scale. If it were, the reader would quickly notice that the curves are quite steep, which can be informative, and sometimes concerning*.

As exemplified by Figure 5, a semi-log representation does not provide more information, also with respect to the steepness of the curves. As we published all our research on antivirals as log-log diagrams e.g. Zirafi et al. ([96]; Zirafi et al., 2015), we would prefer (if possible) to keep the figures in their original log-log representation.

Author response image 1.**DOI:**
http://dx.doi.org/10.7554/eLife.05397.032